# Maximally Useful and Minimally Redundant: The Key to Self Supervised Learning for Imbalanced Data

## Abstract

Contrastive self supervised learning(CSSL) usually makes use of the *multi-view* assumption which states that `all` relevant information must be shared between `all` views. The main objective of CSSL is to maximize the mutual information(MI) between representations of different views and at the same time compress irrelevant information in each representation. Recently, as part of future work, Schwartz Ziv & Yan LeCun pointed out that, when the multi-view assumption is violated, one of the most significant challenges in SSL is in identifying new methods to separate relevant from irrelevant information based on alternative assumptions. Taking a cue from this intuition we make the following contributions in this paper: 1) We develop a CSSL framework wherein multiple images and multiple views(MIMV) are considered as input, which is different from the traditional multi-view assumption 2) We adopt a novel augmentation strategy that includes both normalized (invertible) and augmented (non-invertible) views so that complete information of one image can be preserved and hard augmentation can be chosen for the other image 3) An Information bottleneck(IB) principle is outlined for MIMV to produce optimal representations 4) We introduce a loss function that helps to learn better representations by filtering out extreme features 5) The robustness of our proposed framework is established by applying it to the imbalanced dataset problem wherein we achieve a new state-of-the-art accuracy (2% improvement in Cifar10-LT using Resnet-18, 5% improvement in Cifar100-LT using Resnet-18 and 3% improvement in Imagenet-LT (1k) using Resnet-50).

## 1 Introduction

For image datasets, learning visual representations from unlabeled data is the primary objective of self-supervised learning(SSL) Chen et al. (2020a;c); Khosla et al. (2020). Contrastive self-supervised learning (CSSL) often makes use of a *contrastive loss* function, and its purpose is to spatially converge *similar* instances (that is, maximize similarity between *positive pairs*[1]) and segregate *dissimilar* instances (i.e. minimize similarity between *negative pairs*). CSSL is a special case of multi-view information bottleneck (MVIB) principle, which states that the optimal way to create a useful representation is to maximize the *mutual information*(MI) between the representations of different views while compressing irrelevant information in each representation. This also means that `all` relevant information should be shared between `all` views to maintain semantic consistency.

As noted by Tian et al. (2020b), this semantic consistency may be violated due to the noise introduced by the augmentations, as they tend to overwhelm the shared information. Yet another objective for CSSL is with respect to equating shared information with *useful* information which need not hold true because MI includes low level features (textures, edges), background etc. and these could inflate the MI between representations but do not improve semantic information Poole et al. (2019). Although supervised and unsupervised learning offers more direct access to relevant information, contrastive self-supervised learning is highly dependent on assumptions about the relationship between data and downstream tasks. This reliance makes distinguishing between relevant

---

[1]For example, if we consider images as instances then augmented versions of the same image become positive pairs and different images are considered as negative pairs.

and irrelevant information considerably more challenging, necessitating further assumptions. In a recent work, Ravid Schwart and Yan LeCun Shwartz Ziv & LeCun (2024) pointed out that alternative assumptions need to be developed and new methods for CSSL should be devised that can separate relevant information from irrelevant information.

In this work, we make five contributions: (1) We propose a CSSL framework based on a Multi-Image Multi-View(MIMV) setting that can serve as an alternative for tasks wherein multi-view assumption is violated. 2) We make use of both invertible (normalized) and non-invertible functions (augmented) for transforming views, whereas any traditional self-supervised framework is designed to make use of only non-invertible functions in the form of augmentations; To the best of our knowledge, we are not aware of any CSSL framework that uses a combination of augmented and normalized views to learn visual representations. 3) We make use of two variants of the information bottleneck principle, namely, MVIB (multi-view information bottleneck principle) and Late-MMIB (late-multimodal information bottleneck) so as to extend them to the self supervised Multi-Image Multi-View setting for getting optimal representations. 4) When applied to long-tail datasets, CSSL suffers from the early domination of head classes due to the large number of negatives. In order to address this issue, we introduce a new loss function which helps in eliminating such extreme features that cause the early domination of head classes. 5) We examine the robustness of the MIMV framework by applying it to the dataset imbalance problem, as it is known that CSSL frameworks usually fail in the case of long-tailed learning Jiang et al. (2021); Zhu et al. (2022); Bai et al. (2023). Extensive experimentation with various imbalanced datasets (Cifar10-LT,Cifar100-LT, Imagenet-LT(1k)) and Imagenet-LT subsamples shows a substantive improvement over previous state-of-the-art models( **2%** on Cifar10-LT, **5%** on Cifar100-LT, **3%** on Imagenet-LT(1k)).

## 2 RELATED WORK

CSSL falls into the family of Multi-view Self supervised learning(MVSSL) which in turn is classified into three families, viz., contrastive Chen et al. (2020a;c); He et al. (2020); Caron et al. (2021); Bardes et al. (2022b), clustering Caron et al. (2018; 2020) and distillation-based Grill et al. (2020). The proposed work belongs to the contrastive family van den Oord et al. (2018); Chen et al. (2020a;b); He et al. (2020); Chen et al. (2020c); Tian et al. (2020a); Chuang et al. (2020); Khosla et al. (2020); Chen et al. (2021); Jiang et al. (2021). The basic assumption in CSSL is to generate two or more views(multi-view) for each data sample by using augmentations van den Oord et al. (2018); Tian et al. (2020b) so that the semantic information shared between the views remains as intact as that of the original sample. To the best of our knowledge, none of the CSSL methods outlined above address multi image multi-view perspective as proposed in this work. Moreover, the views in CSSL are generated through augmentations which are functions that are invertible in nature, whereas in our case a combination of invertible and non-invertible functions is made use of.

All major CSSL frameworks are supported by the Multi-view Information bottleneck principle(MVIB) Tishby et al. (2000); Sridharan & Kakade (2008); Tishby & Zaslavsky (2015); Federici et al. (2020); Tsai et al. (2021); Gálvez et al. (2023); Wang et al. (2023); Louizos et al. (2024), which aims to learn representations that maximize *shared/relevant* information across views of the same sample while minimizing *unnecessary/redundant* information. The traditional MVIB principle is designed for a single image multi-view perspective which may not work in scenarios wherein multiple images and multiple views are involved as in our case. To this end, we introduce the MIMV bottleneck principle.

Recent work on SSL has demonstrated that compared to fully supervised models, architectures that leverage self-supervised pretraining are more resistant to class imbalance Yang & Xu (2020); Jiang et al. (2021); Liu et al. (2022); Lin et al. (2023); Bai et al. (2023); Kukleva et al. (2023). These methods advocate using out-of-distribution (OOD) data or in-domain (ID) data samples to balance the minority class to boost the long-tailed learning performance of SSL. Most of the methods mentioned above Yang & Xu (2020); Liu et al. (2022); Lin et al. (2023); Bai et al. (2023); Kukleva et al. (2023) make use of the additional data (ID or OOD) to re-balance the features. Other methods address the long-tailed problem by strengthening minority features through sampling or reweighting techniques Liu et al. (2022). There is also a work that addresses the issue of data set imbalance in SSL using a prototypical re-balancing strategy Lin et al. (2023). Of these, except for Bai et al. (2023), all other works address the problem of data imbalance by sampling with extra domain data

that can re-balance the minority class. CL by nature is biased towards the head classes due to the large number of negatives. We introduce a contrastive loss function, which takes advantage of multi image multi view design and helps in minimizing the dominance of the head classes.

# 3 PRELIMINARIES AND FRAMEWORK

Most SSL frameworks He et al. (2020); Khosla et al. (2020); Jiang et al. (2021); Ren et al. (2022) make use of the NT-Xent loss or its variants (Our analysis of NT-Xent loss will be with respect to the SIMCLR Chen et al. (2020a) framework which falls under the family of multiview self-supervised learning(MVSSL)). NT-Xent loss computes the pairwise similarity between two augmented views of an image by making use of the cosine similarity as given in Equation ( 1).

$$\mathcal{L}_{i,j} = -log \frac{exp(sim(z_i, z_j)/\tau)}{\sum_{k=1}^{n} \mathbb{1}_{\{i \neq k\}} exp(sim(z_i, z_k)/\tau)} \tag{1}$$

Here, $z_i$ and $z_j$ are two latent representations of the augmented views, potentially from a single image or other images. $z_k$ is from the remaining latent representation of augmented pairs, excluding $(z_i, z_j)$. Suppose that $X_1 \in \mathbb{D}$ is an image, $x_1^{a_1}, x_1^{a_2} \leftarrow X_1$ are two augmentations, and $z_1^{a_1}, z_2^{a_2} \leftarrow x_1^{a_1}, x_1^{a_2}$ are the latent representations of the augmented views. In this example, NT-Xent needs to calculate the similarity of four pairs $(z_1^{a_1}, z_1^{a_1})$, $(z_1^{a_1}, z_1^{a_2})$, $(z_1^{a_2}, z_1^{a_1})$, $(z_1^{a_2}, z_1^{a_2})$ for a given image. Of these given pairs, $(z_1^{a_1} \cdot z_1^{a_1})$ and $(z_1^{a_2} \cdot z_1^{a_2})$ are those pairs that are similar to itself (which means $z_1^{a_1} \cdot z_1^{a_1} = 1 \ and \ z_1^{a_2} \cdot z_1^{a_2} = 1$) and therefore these pairs are eliminated by NT-Xent. By the symmetric property of the vector dot product, $(sim(z_1^{a_1}, z_1^{a_2}) = sim(z_1^{a_2}, z_1^{a_1}))$, the only pair that is important is $sim(z_1^{a_1}, z_2^{a_2}) = z_1^{a_1} \cdot z_2^{a_2}$. The final loss is calculated as given in Equation:1. Similarly, if we start with two images, as in our case, for the loss calculation, NT-Xent would generate sixteen pairs of which only a few are relevant, as shown in Figure:1. It can be visualized from the figure that the upper triangular matrix has the same similarities as the lower triangular matrix. So to calculate the similarity pairs that are relevant, only one of these matrices needs to be taken into account. Among these pairs, those having instances from the same source (green) as well as those having instances from different sources (pink, red) are identified for loss calculation in SIMCLR.

MIMV assumes that there are multiple images to start with, which results in a larger number of pair formations than that of SIMCLR. It is necessary for our proposed framework to have a much more compact representation so that the focus will be on the similarity between *compact representations* rather than on finding the similarity between *representations of augmented views*. We take inspiration from the work of Li et al. (2018) and show that of the six resulting pairs (in this paper, we term them as *Intra Similarity* and *Inter Similarity* between representations), only three combinations $((z_1^{a_1} \circledast z_2^{a_2}).(z_1^{a_2} \circledast z_2^{a_1}), (z_1^{a_1} \circledast z_2^{a_1}).(z_1^{a_2} \circledast z_2^{a_2}), (z_1^{a_1} \circledast z_1^{a_2}).(z_2^{a_1} \circledast z_2^{a_2}))$ are possible by using `fusion` representation Li et al. (2018). Fusion representation is primarily used to get a combined neural network representation for multi-

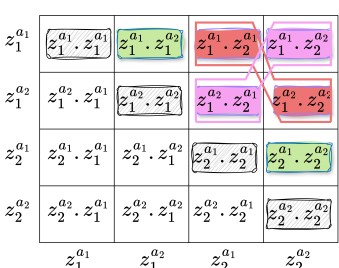

Figure 1: Pair formation in SimCLR with two images

modal (for instance, image, text, audio, etc.) learning Karpathy & Fei-Fei (2015); Kiela & Bottou (2014); McLaughlin et al. (2016).

For the MIMV framework, we start with two images $X_1, X_2 \in \mathbb{D}$ that are drawn at random. $x_1^{a_1}$ and $x_1^{a_2}$ are the two augmented views of $X_1$ and $x_2^{a_1}$ and $x_2^{a_2}$ are the two augmented views of $X_2$. We define $z_1^{a_1}, z_1^{a_2}$ are the two representations of $x_1^{a_1}$ and $x_1^{a_2}$ and $z_2^{a_1}, z_2^{a_2}$ are the two representations of $x_2^{a_1}$ and $x_2^{a_2}$. We then have six unique pairwise cosine similarities between the representations which help in decision making, as shown in Figure: 1. From these six pairs, our aim is to generate possible combinations to simulate a representation space that supports the MIMV approach. As pointed out in the introduction, we identify that $z_1^{a_1}.z_1^{a_2}$ and $z_2^{a_1}.z_2^{a_2}$ are *intra* similarity and $(z_1^{a_1}.z_2^{a_1}), (z_1^{a_1}.z_2^{a_2}), (z_1^{a_2}.z_2^{a_1}), (z_1^{a_2}.z_2^{a_2})$ are the *Inter Similarity* pairs between representations. The pairwise intra similarity is denoted in green and the two pairwise inter-similarity is denoted in red/pink as given in Figure 1. If we follow the SIMCLR framework, we need to compute the

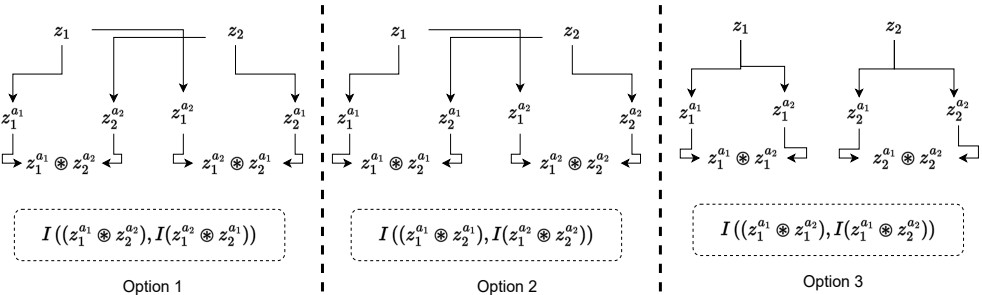

Figure 2: Multi-Image Multi-View analysis

similarity between all pairs of representations. What we ideally need is to fuse the data from multiple representations into a single representation to maximize the *mutual information* between the representations ($I(z_1, z_2; y)$).This will also help in retaining more relevant information than any individual representation $I(z_1; y)$ *or* $I(z_2; y)$. In information-theoretic terms we can give it as follows;

$$I(z_1, z_2; y) \geq max(I(z_1; y), I(z_2; y)) \tag{2}$$

By using the data processing inequality principle Cover & Thomas (2006), the above statement can be rewritten as

$$I(z_1, z_2; y) \geq I(f(z_1, z_2); y) \geq max(I(z_1; y), I(z_2; y)) \tag{3}$$

Let $f(z_1, z_2) = z_1 \circledast z_2$. Here we use $f(.)$ as a fusion function Li et al. (2018), that takes two representations and applies $\circledast$ as a fusion operation. In this paper $\circledast$ is interpreted in two ways as follows:

$$f(z_1, z_2) = z_1 + z_2 \ (sum) \quad or \quad f(z_1, z_2) = [z_1, z_2] \ (concatenate)$$

The idea is to form a more compact representation space by making use of a fusion function as defined above. Using the pairwise inter and intra similarity as shown in Figure 2, we can form at most three unique compact representations, which we term as Option-1, Option-2, and Option-3. A sketch of how we arrive at Option-1 is given below in ( 4), and since the procedure remains the same for the other two options, we only give the final representations of Option-2 and Option-3. Details can be found in Appendix. *Option-1:* In this, we select pairwise intra similarity(green) $(z_1^{a_1}.z_1^{a_2})$ , $(z_2^{a_2}.z_2^{a_2})$ and pairwise inter similarity(red) $(z_1^{a_1}.z_2^{a_1})$ , $(z_2^{a_2}, z_2^{a_2})$.

$$=> f\left((z_1^{a_1}.z_1^{a_2}), (z_1^{a_1}.z_2^{a_1}), (z_1^{a_2}.z_2^{a_2}), (z_2^{a_1}, z_2^{a_2})\right)$$
$$=> (z_1^{a_1}.z_1^{a_2}) \circledast (z_1^{a_1}.z_2^{a_1}) \circledast (z_1^{a_2}.z_2^{a_2}) \circledast (z_2^{a_1}, z_2^{a_2})$$
$$=> z_1^{a_1}.(z_1^{a_2} \circledast z_2^{a_1}) \circledast z_2^{a_2}.(z_1^{a_2} \circledast z_2^{a_1})$$
$$=> (z_1^{a_1} \circledast z_2^{a_2}).(z_1^{a_2} \circledast z_2^{a_1}) \tag{4}$$

The other two resulting compact representations can be given as follows: *Option-2:*$(z_1^{a_1} \circledast z_2^{a_1}).(z_1^{a_2} \circledast z_2^{a_2})$ *Option-3:*$(z_1^{a_1} \circledast z_1^{a_2}).(z_2^{a_1} \circledast z_2^{a_2})$

### 3.1 SHARED INFORMATION ANALYSIS:

Taking inspiration from the data processing inequality principle Cover & Thomas (2006)($I(x; y) \geq I(v; y)$), we make use of both invertible (normalized) and non-invertible (augmented) functions for transforming views(v) whereas any traditional self-supervised framework is designed to make use of only non-invertible functions in the form of augmentations. We initially carried out experiments to find the sanctity of this intuition. The experimental results are outlined in Figure 4 which shows that the addition of the normalized view along with augmentation alone improves SIMCLR performance by 2. 8% over the test data.

Let $X_A, X_C \in \mathbb{D}$ be drawn at random, where image $X_A$ is an anchor image and $X_C$ is considered as its counterpart. Further, $x_A^n, x_C^n$ and $x_A^a, x_C^a$ are the normalized and augmented versions of anchor

$X_A$ and its counterpart $X_C$, respectively. $z_A^n, z_C^n$ and $z_A^a, z_C^a$ are the latent representations of the normalized and augmented versions of anchor $X_A$ and its counterpart $X_C$, respectively. Let the information volume $V$ be the measurement of the anchor-related information contained within a representation. Volume $V = 1$ ($max$), when the representation retains complete information as passed from its view, while $V = 0$ ($min$), when there is complete loss of information. Other information volume lies between 0 and 1.($min$) $0 \leq \theta(z) \leq 1$ ($max$). $\theta$ is measuring the volume and $z$ is an instance

$$\text{Information Ratio } (I_R(z)) = \frac{\text{representation volume}}{\text{total volume}} = \frac{\theta(z)}{V}$$

Using Equation: $(I(x; y) \geq I(z; y))$, we get $V(z_A^n) = 1$, $V(z_C^n) = 1$. The reason is that we are considering the normalized version of an image for both anchor and it's counterpart. The normalized version of an image is invertible in nature, and therefore there is no loss of information. On the other hand, an augmented version of both images might lose information because of its noninvertible nature. To make things clear, $let\ V(z_A^a) = 0.8$, $V(z_C^a) = 0.8$. Then the total information V for a pair of $(z_A, z_C)$ is measured by $V(z_A \circledast z_C) = V(z_A) + V(z_C)$. This analysis is needed to measure the shared information between pairs of representations as shown in Figure: 2.

$$Option\ 1: \quad (z_A^n \circledast z_C^a), (z_A^a \circledast z_C^n)$$

$$I_R(A) = \frac{\theta(A)}{V} = \frac{\theta(A)}{V(z_A^n) + V(z_C^a)}, \frac{\theta(A)}{V(z_A^a) + V(z_C^n)}$$

$$I_R(A) = \frac{1}{1 + 0.8}, \frac{0.8}{0.8 + 1} = 0.55, 0.44$$

Both pairs have a volume of anchor-related information $0.55, 0.44$. A maximum shared information volume between our pairs can be $0.44$. Likewise, we can calculate the shared information of the remaining two pairs. *Option 2:* Both pairs have $0.5, 0.5$ anchor-related information volume. A maximum shared information volume between our pairs can be $0.5$. *Option 3:* Both pairs have a volume of anchor-related information $1, 0$. A maximum shared information volume between our pairs can be $0$. This is one way to calculate the shared information as well as the

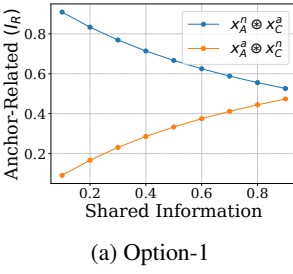
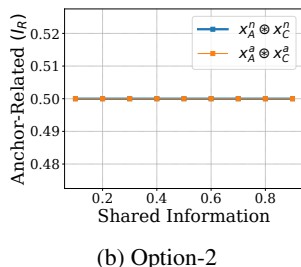
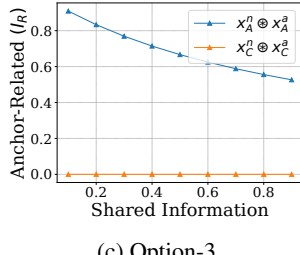

(a) Option-1  (b) Option-2  (c) Option-3

Figure 3: MIMV options

anchor-related information held by each compact representation. Detailed experimental results are given in Figure 3.This figure shows the anchor-related information (Y-axis) held by a pair with respect to information loss (X-axis). Shared information can be common information between pairs.

As information loss decreases, the shared information increases in Figure 3a. Figure 3b does not reflect any change in shared information. Figure 3c has one bad representation that has no information on the anchor. Using this analysis, we find that the representation pair in figure 3a is the most promising pair to go with.

MIMV alone with augmented views outperforms SIM-CLR by a margin of 3.5% over test data. Furthermore, by combining MIMV with normalized and augmented views, an improvement of more than 8% is achieved over the test data as shown in Figure: 4. All experiments were carried out on Cifar100-LT with 2000 epochs. These results motivated us to further investigate our experimental results from a theoretical perspective, the details of which are outlined in the following sections.

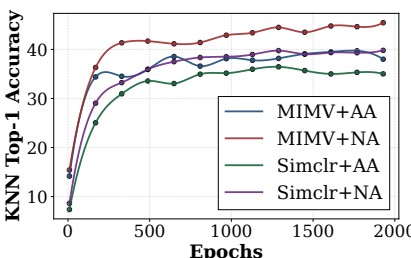

Figure 4: Simclr and MIMV analysis (NA-Normalized, Augmented pair), (AA-Augmented, Augmented pair)

## 3.2 OUR OBJECTIVE:

From the analysis given above, one can see that we have narrowed our focus to a single pair (Figure 3a) $(z_A^n \circledast z_C^n), (z_C^a \circledast z_A^a)$. A deeper analysis of this unique pair is needed to build the final design of our MIMV framework. This can be achieved through a two-stage process. At the first stage, we observe that the final resulting pair has a joint representation (fusion) of multiple modalities(multiple images). Since the fusion happens at the representation level we can relate this with the late multi-modal information bottleneck principle (Late-MMIB) Mai et al. (2022) in which each modality is encoded independently and fusion happens at the representation level. The objective of Late- MMIB is as follows:

$$max\ I(z_1; z_2) - \beta_1 I(z_1; x_1) - \beta_2 I(z_2; x_2) \tag{5}$$

The goal is to extract complementary, task-relevant representations from each modality and fuse them in a way that maximizes predictive performance while minimizing redundancy. In the second stage, we could visualize our fusion representation from a multiview representation learning (MVRL) Wan et al. (2021) perspective and can use $(z_A^n \circledast z_C^n)$ and $(z_C^a \circledast z_A^a)$ as two different representations. Multiview representation learning Hjelm et al. (2019); Tian et al. (2020b); Federici et al. (2020) relies on the redundancy of multiple views from the same source, and the objective is to obtain a compressed representation per view while maximizing the shared information between representations (multiview information bottleneck principle(MVIB)) Tishby & Zaslavsky (2015); Tishby et al. (2000); Wan et al. (2021).

$$max\ I(z_1; z_2) - \beta(I(z_1; x_1) + I(z_2; x_2)) \tag{6}$$

Here, $x_1$ and $x_2$ are two augmented views of $X \in \mathbb{D}$ while $z_1$ and $z_2$ are the two representations of $x_1$ and $x_2$. $\beta$ is a trade-off parameter that controls compression. From the Late-MMIB and MVIB principles as outlined above, we derive our formulation based on the final useful pair.

$$I((z_A^n, z_C^a); (z_A^a, z_C^n))$$
$$-\beta[I(z_A^n; z_C^a) - \beta_1 I(z_A^n; x_A^n) - \beta_2 I(z_C^a; x_C^a) + I(z_A^a; z_C^n) - \beta_1 I(z_A^a; x_A^a) - \beta_2 I(z_C^n; x_C^n)] \tag{7}$$

Here, $I(z_A^n; z_C^a)$ and $I(z_A^a; z_C^n)$ are complementary representations which ensure `minimal redundancy` as well as preservation of task-related information, On the other hand, $I((z_A^n, z_C^a), (z_A^a, z_C^n))$ maximizes the mutual information between these complementary pairs to ensure `maximal usefulness`. $\beta, \beta_1$ and $\beta_2$ are trade-off parameters to take care of compression. We take advantage of the loss of NT-Xent as a surrogate to optimize the MIMV objective (Equation: 7), which includes explicit compression terms such as $\beta, \beta_1$ and $\beta_2$. However, NT-Xent absorbs these terms through architectural and training design choices such as augmentations, projection heads, and temperature scaling, effectively controlling the trade-off without requiring explicit $\beta$ terms. We can evaluate the mutual information between the representations $I(z_A^n, z_C^a)$ and $I(z_A^a, z_C^n)$ as follows:

$$I((z_A^n, z_C^a); (z_A^a, z_C^n)) \geq \log(N) - L_{MIMV} + c \tag{8}$$

Minimizing this NT-Xent loss $L_{MIMV}$ maximizes mutual information. More details can be found in the appendix. The analysis given above clearly demonstrates that the resulting pairs as shown in Figure 2 are the most crucial in an MIMV setting and also meet the criteria of ***being maximally useful and minimally redundant***. Based on these observations, we propose a framework for the MIMV objective as shown in Figure: 5.

## 3.3 PRETRAINING WITH MOMENTUM LEARNING:

In Figure: 5, we start with an anchor image $(X_A)$ and it's counterpart $X_C$. Further, $x_A^n, x_C^n$ and $x_A^a, x_C^a$ are the normalized and augmented versions of anchor $X_A$ and it's counterpart $X_C$ respectively. We adopted the momentum learning He et al. (2020) paradigm for pretraining as it has been the preferred choice of various self-supervised frameworks Chen et al. (2020c); Grill et al. (2020); Caron et al. (2020); Chen & He (2021). Momentum learning uses two parallel networks simultaneously for pretraining, known as the online encoder and the target encoder. In the figure, we denote the online encoder as $encoder_q$ and the target encoder as $encoder_k$. The online encoder representations are denoted as $z_A^c = encoder_q(x_A^n)$ and $z_C^a = encoder_q(x_C^n)$. On the other hand, target encoder representations are denoted as $z_A^a = encoder_k(x_A^a)$ and $z_C^n = encoder_k(x_C^n)$. $encoder_k$ is exponential moving average(EMA) of $encoder_q$'s parameters, while $encoder_q$ is used to update the gradients by backpropagation.

---

**Algorithm 1** MIMV:

  **Input:** Batch size N, Normalized Images $x_A^n, x_C^n$, Augmented Image $x_A^a, x_C^a$, $encoder_q$, $encoder_k$,

  **for** batch in train˙loader **do**

    $x_A^n, x_n^C, x_A^a, x_C^a$ = batch

    $z_A^n, z_C^a = encoder_q(x_A^n), encoder_q(x_C^a)$

    **with** no˙grad():

      $momentum\_update\_encoder_k()$

      $z_a^A, z_C^n = encoder_k(x_A^a), encoder_k(x_C^n)$

    $S = sim\left((z_A^n \circledast z_C^a), (z_A^a \circledast z_C^n)\right)$

    $S(i,j) = \{S_{i,j}.\mathbb{1}_{[\lambda_l, \lambda_h]} S_{i,j}\}$

    $\mathcal{L} = -log\left[\frac{e^{S/\tau}}{\sum_{k=1}^{2N} \mathbb{1}_{\{i \neq k\}} e^{S/\tau}}\right]$

  **end for**

  return $\mathcal{L}$

---

The overall loss over these representation pairs can be computed as

$$\mathcal{L}_{i,j} = -log \frac{e^{\left(sim((z_A^n \circledast z_C^a)_i \ (z_A^a \circledast z_C^n)_j)/\tau\right)}}{\sum_{k=1}^{2N} \mathbb{1}_{\{i \neq k\}} e^{\left(sim\left((z_A^n \circledast z_C^a)_i \ (z_A^a \circledast z_C^n)_k\right)/\tau\right)}} \quad (9)$$

This loss function is a variant of the NT-Xent 1 loss function, which adopts a compact representation to form the positives and negatives. The loss function given above is modified to eliminate extreme features that are within a certain threshold ($\lambda_l$ (lower limit), $\lambda_h$ (higher limit)). Two extreme cases that could result are when we have extremely similar/dissimilar images. Let $z_A$ and $z_C$ be almost similar image representations such that the similarity between both representations is close to one ($sim(z_A, z_C) \simeq 1$). In this case of extreme similarity, intra similarity ($z_A^n.z_A^a$ $and$ $z_C^n.z_C^a$) becomes irrelevant. It can be high or low, but it does not make any difference because the inter similarity $z_A^n.z_C^n$ $and$ $z_C^a.z_A^a$ already dominates.

$$\text{if } z_A^n.z_A^a \circledast z_A^n.z_C^n \circledast z_C^a.z_A^a \circledast z_C^a.z_C^n \simeq 1$$

These types of cases can be ignored because good examples are those that exist closer in latent space but are

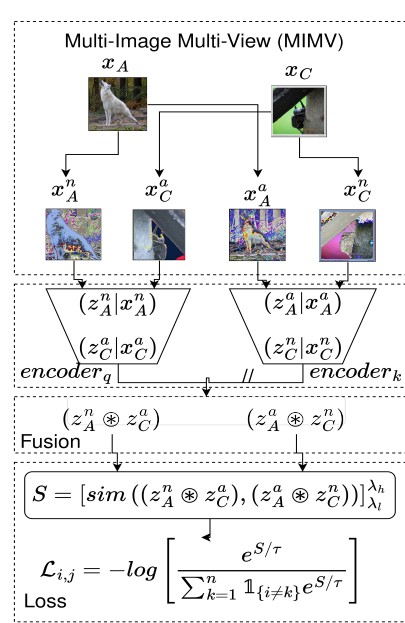

Figure 5: MIMV illustration

symmetrically different. This is not possible in extreme cases. Similarly, if we have extremely dissimilar views, i.e. $z_A$ and $z_C$ are completely different from each other, then $sim(z_A, z_C) \simeq 0$. Here $(z_A^n.z_C^a)$ $and$ $(z_A^a.z_C^n)$ are inter similarity pairs. Since $z_A$ and $z_C$ are completely distinct from each other, we end up with

$$(z_A^n.z_C^a) \simeq 0 \ and \ (z_A^a.z_C^n) \simeq 0$$

and therefore in these kinds of scenario, decision making is highly dependent on intra-similarity pairs $x_i.x_i'$ $and$ $z_C.z_C'$. Even though these pairs are instances from the same image, their similarity may decrease because of more challenging augmentation of the images and may result in extreme case as follows:

$$z_A^n.z_A^a \circledast z_A^n.z_C^n \circledast z_C^a.z_A^a \circledast z_C^a.z_C^n \simeq 0$$

This is a case of extreme dissimilarity and these type of cases are also not important and should be ignored. The proposed loss function given in Equation 10 takes care of eliminating these extreme cases given a certain threshold. The modified loss function can be given as follows:

$$\text{Let } S = Sim\left((z_A^n \circledast z_C^a).(z_A^a \circledast z_C^n)\right)$$

$$S(i,j) = \{S_{i,j}.1_{[\lambda_l, \lambda_h]}S_{i,j}\}$$

$$\text{where } 1_{[\lambda_l, \lambda_h]}S_{i,j} = \begin{cases} \mathcal{S}_{i,j}, & \text{if } \lambda_l \leq S_{i,j} \leq \lambda_h \\ 0, & \text{otherwise} \end{cases}$$

$$\text{Final Loss: } \mathcal{L}_{i,j} = -log\left[\frac{e^{S/\tau}}{\sum_{k=1}^{2N} 1_{\{i \neq k\}} e^{S/\tau}}\right] \quad (10)$$

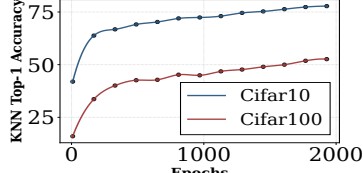

Figure 6: Knn Top-1 Accuracy on Cifar10 and Cifar100

The loss function in Equation 10 can be explained as follows: First, we compute the pairwise similarity between our aggregated representations. Further, using a certain threshold, extreme similarities are eliminated. Finally, these refined similarities are fed into our final loss function. This new loss function becomes more robust as the training progresses as it is able to eliminate more extreme features and this characteristic of the loss function is shown in Figures: 6 and 7. The detailed algorithm related to our Multi-Image Multi-View (MIMV) approach is given in algorithm 1.

## 4 RESULTS AND DISCUSSION

We use exponential distribution to create Cifar10-LT Krizhevsky et al. (2009) and Cifar100-LT Krizhevsky et al. (2009) with an imbalance factor of $r = 0.01$, taken from Liu et al. (2022). For ImageNet, we used a Pareto distribution to create ImageNet-LT Russakovsky et al. (2015). Pareto distribution is more likely to generate real-world data. We followed the ImageNet-LT and their subset construction suggested by Lie et al. Liu et al. (2022). An imbalance factor of $\alpha = 0.004$ produces a challenging imbalance dataset. Furthermore, we used subsampling to make some more challenging subsets of the dataset, while its structure remains intact. We use SubsetRandom-Sampler with the Stratified Sampling technique to create samples.

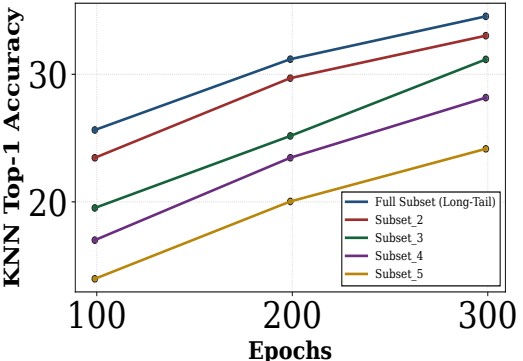

Figure 7: KNN Top-1 Accuracy on ImageNet(1K) dataset

**Pretraining** To underline the precision of our approach, we present and validate our results on the ImageNet dataset. We used Resent18 He et al. (2016) and Resnet50 He et al. (2016) as our backbone architectures. We employed a three-layer MLP consisting of three fully connected layers with three batch normalization units and two RELU activations for the projection head. We used a batch size of 1024 for Cifar10-LT and Cifar100-LT, while a batch size of 256 was used for Imagenet-LT. We employ stochastic gradient descent with a learning rate of 3.0 for Cifar10-LT and Cifar100-LT, while 0.5 for Imagenet-LT subsamples made by sampling factor $s = 0.125, 0.25, 0.50$ and 0.1 for remaining subsamples $s = 0.75, 1.0$. We used a cosine decay learning rate with a warming of 10% of the max epochs with a weight decay of $1e-4$ to learn a more robust representation. To update the weights of the target network ($encoder_k$), we used a momentum of 0.9. All metrics reported here are trained with 300 epochs in Imagenet-LT and 2000 epochs in Cifar10-LT and Cifar100-LT, and an average of three runs. To eliminate extreme features by the limit of $(low)$ $\lambda_l, \lambda_h$ $(high)$, we use $\lambda_l = 0.1$ $and$ $\lambda_h = 0.9$ for Cifar10-LT and Cifar100-LT. For Imagenet-LT and their subsets, we use $\lambda_l = 0.1$ $and$ $\lambda_h = 1.0$. We observed that we must carefully use this limit as it may result in numerical instability. We noticed that $\lambda_h$ is more sensitive to numerical instability.

**KNN Evaluation** To compute this KNN evaluation of the trained features, we use K=200. To achieve KNN accuracy, we first extract the features of training data from our trained $encoder_q$. We also consider the labels of the training data. With the features and their respective labels, we prepared a set of feature bank. On the other hand, we collect features for the test data and then feed

them to predict their respective class. Finally, to calculate the accuracy, we compare these predicted labels with the actual test labels. The experimental results of KNN are given in Figure: 6, 7

**Linear Evaluation** We demonstrate the model's versatility by freezing the model and letting the linear layer learn for our downstream task. The linear layer, with dimensions of (feature size, number of classes), is a testament to this adaptability. We adopted cosine learning weight decay without warm-up with $10\%$ of the max epochs and a learning rate of $0.005$ for this task. Detailed results of the linear evaluation of various methods are given in Table 1, 2, 3.

Table 1: Cifar10-LT (exponential) Imbalance Dataset Results(Resnet-18)

| Model | Imbalance Type - exp | | | | |
| --- | --- | --- | --- | --- | --- |
| | Acc↑ | Frequent↑ | Medium↑ | Rare↑ | Std↓ |
| **MoCoV2** He et al. (2020) | 74.76 | 80.70 | 74.36 | 69.8 | 4.46 |
| **Byol** Grill et al. (2020) | 75.66 | 81.70 | 69.83 | 75.43 | 4.85 |
| **SwAV** Caron et al. (2020) | 76.60 | 85.40 | 71.10 | 73.30 | 6.29 |
| **VicReg** Bardes et al. (2022a) | 73.32 | 76.26 | 74.46 | 70.27 | 2.51 |
| **SimCLR** Chen et al. (2020a) | 76.77 | 82.16 | 76.9 | 71.20 | 4.47 |
| **SDCLR** Jiang et al. (2021) | 80.49 | 88.30 | 78.07 | 75.10 | 5.66 |
| **FASSL** Lin et al. (2023) | 80.69 | 86.55 | 76.30 | 78.80 | 4.23 |
| **SimSiam** Ren et al. (2022) | 81.40 | - | - | - | - |
| **Ours(Sum)** | **83.66** | **87.30** | **84.36** | **80.40** | **2.82** |

Table 2: Cifar100-LT (exponential) Imbalance Dataset Results(Resnet-18)

| Model | Imbalance Type - exp | | | | |
| --- | --- | --- | --- | --- | --- |
| | Acc↑ | Frequent↑ | Medium↑ | Rare↑ | Std↓ |
| **SimCLR** Chen et al. (2020a) | 44.85 | 47.81 | 41.48 | 44.17 | 2.59 |
| **MoCoV2** He et al. (2020) | 46.37 | - | - | - | - |
| **Byol** Grill et al. (2020) | 47.00 | 53.62 | 47.42 | 45.55 | 3.45 |
| **VicReg** Bardes et al. (2022a) | 45.26 | 48.21 | 43.27 | 44.35 | 2.12 |
| **SwAV** Caron et al. (2020) | 48.86 | 49.97 | 47.03 | 44.00 | 2.44 |
| **BCL-I** Zhou et al. (2022) | 52.22 | 55.35 | 53.03 | 48.27 | 2.95 |
| **SDCLR** Jiang et al. (2021) | 54.94 | 58.79 | 55.03 | 51.00 | 3.18 |
| **FASSL** Lin et al. (2023) | 55.27 | 57.74 | 54.52 | 53.55 | **1.79** |
| **Ours(Concatenate)** | **58.89** | **61.54** | **56.18** | **58.94** | 2.19 |
| **Ours(Sum)** | **60.18** | **62.24** | **56.75** | **61.50** | 2.42 |

Table 3: Imagenet-LT subsamples results (Resnet-50)

| Model | Subsampling Ratio (s) | | | | |
| --- | --- | --- | --- | --- | --- |
| | s=1 | s=0.75 | s=0.50 | s=0.25 | s=0.125 |
| **SimCLR** Chen et al. (2020a) | 46.81 | - | - | - | - |
| **MoCoV2** He et al. (2020) | 49.50 | 43.20 | 39.5 | 36.60 | 30.50 |
| **SDCLR** Jiang et al. (2021) | 46.62 | - | - | - | - |
| **Byol** Grill et al. (2020) | 43.16 | - | - | - | - |
| **VicReg** Bardes et al. (2022a) | 38.24 | - | - | - | - |
| **Ours(Sum)** | **52.90** | **50.25** | **49.04** | **45.51** | **39.68** |

## 5 CONCLUSIONS AND FUTURE WORK

We propose a *Multi-Image Multi-View* approach (MIMV) in contrastive self-supervised learning (CSSL), which differs from the traditional multi-view setup in both theory and practice. Rather than considering multiple views of a single image, we start with two images and study the formation of similarity pairs, including inter-/intra-discriminatory pairs. In order to have a compact representation of the generated pairs, we make use of the fusion representation, which is often used as a tool to fuse multiple modalities in multimodal representation learning. A theoretical study is carried out to search the space of all possible pairs to identify the useful ones. Based on the information

shared between pairs, we identified the most useful pair. To develop a framework that will support only useful pairs having similar structural patterns, we adopted the principles of multi modal information bottleneck (MMIB) as well as multi-view information bottleneck (MVIB). From these two principles, we derived our formulation of MIMV information bottleneck principle (MIMVIB). We developed a framework using momentum learning with MIMVIB, in which the typical pattern of using two augmented views of a single image in CSSL is replaced with that of one augmented view and one normalized view. In addition, we improved our proposed model by eliminating extreme features to obtain a more robust representation. We evaluated the proposed model on various imbalanced datasets (Cifar10-LT, Cifar100-LT, Imagenet-LT(1K)) and achieved state-of-the-art results. Although we have few promising results related to our framework on standard balanced datasets for self-supervised learning, we have not carried out extensive experimentation in this regard. This could be a future direction to follow.

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
