# A  Supplementary Material

## A.1  Notations

| | | |
|---|---|---|
| Dataset | - | $\mathcal{D}$ |
| Latent Space | - | $\mathbb{R}$ |
| Input | - | x |
| label | - | y |
| views | - | v |
| views from anchor image | - | $v_A^n, v_A^a, \ldots$ |
| views from counterpart image | - | $v_C^n, v_C^a, \ldots$ |
| Latent Representations | - | $z_A^n, z_C^a, \ldots$ |
| Fusion Operation | - | $\circledast$ |

| | | |
|---|---|---|
| $x_1, x_2$ | $\in \mathcal{D}$ , | where $x_1$ and $x_2$ are images drawn from dataset $\mathcal{D}$. |
| $x_1^{a_1}, x_1^{a_2}$ | $\in x_1,$ | |
| $x_2^{a_1}, x_2^{a_2}$ | $\in x_2,$ | |
| $z_1^{a_1}, z_1^{a_2}$ | $\in x_1,$ | |
| $z_2^{a_1}, z_2^{a_2}$ | $\in x_2,$ | |
| X,Y | $\in \mathcal{D}$ , | where X is an image and Y is label. |
| $x_A, x_C$ | $\in$ X, | |
| $y_A, y_C$ | $\in$ Y, | |
| $v_A^n, v_C^n$ | $\leftarrow x_A, x_C,$ | Normalized version of images. |
| $v_A^a, v_C^a$ | $\leftarrow x_A, x_C,$ | Augmented version of images. |
| $z_A^n, z_A^a$ | $\leftarrow v_A^n, v_A^a,$ | Vector representation of anchor image. |
| $z_C^n, z_C^a$ | $\leftarrow v_C^n, v_C^a,$ | Vector representation of counterpart image. |

## A.2  From SIMCLR representation space to Multi-Image Multi-View(MIMV) representation

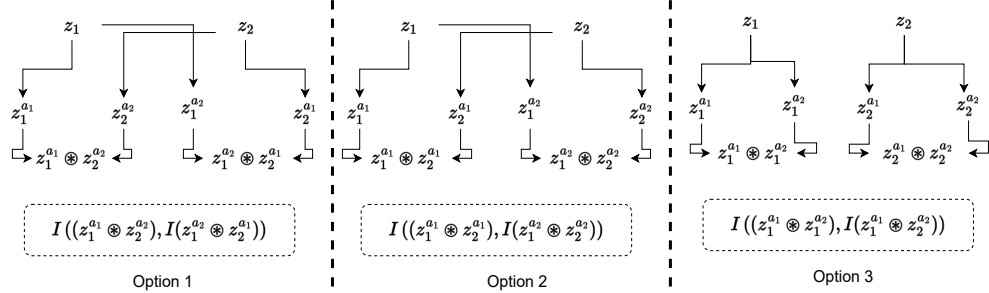

Figure 1: Multi-Image Multi-View (MIMV) analysis

From Figure 22, $x_1, x_2 \in \mathcal{D}$ are two images drawn at random. $x_1^{a_1}, x_1^{a_2} \in x_1$ and $x_2^{a_1}, x_2^{a_2} \in x_2$ are two augmented views from each image. $z_1^{a_1}, z_1^{a_2} \in x_1$ $z_2^{a_1}, z_2^{a_2} \in x_2$ are their image representations.

**Option-1:** In this, we select pairwise intra similarity(green) $(z_1^{a_1}.z_1^{a_2})$ , $(z_2^{a_1}.z_2^{a_2})$ and pairwise inter similarity(red) $(z_1^{a_1}.z_2^{a_1})$ , $(z_2^{a_2}, z_2^{a_2})$.

$$=> f\left((z_1^{a_1}.z_1^{a_2}) , (z_1^{a_1}.z_2^{a_1}) , (z_1^{a_2}.z_2^{a_2}) , (z_2^{a_1}, z_2^{a_2})\right)$$
$$=> (z_1^{a_1}.z_1^{a_2}) \circledast (z_1^{a_1}.z_2^{a_1}) \circledast (z_1^{a_2}.z_2^{a_2}) \circledast (z_2^{a_1}, z_2^{a_2})$$
$$=> z_1^{a_1}.(z_1^{a_2} \circledast z_2^{a_1}) \circledast z_2^{a_2}.(z_1^{a_2} \circledast z_2^{a_1})$$
$$=> (z_1^{a_1} \circledast z_2^{a_2}).(z_1^{a_2} \circledast z_2^{a_1}) \tag{1}$$

**Option 2:** In this, we select one pairwise intra(green) similarity $(z_1^{a_1}.z_1^{a_2})$ , $(z_2^{a_1}.z_2^{a_2})$ and one pairwise inter(pink) similarity $(z_1^{a_2}.z_2^{a_1})$ , $(z_1^{a_1}, z_2^{a_2})$ .

$$=> (z_1^{a_1}.z_1^{a_2}) \circledast (z_1^{a_1}.z_2^{a_2}) \circledast (z_1^{a_2}.z_2^{a_1}) \circledast (z_2^{a_1}.z_2^{a_2})$$
$$=> z_1^{a_1}.(z_1^{a_2} \circledast z_2^{a_2}) \circledast z_2^{a_1}.(z_1^{a_2} \circledast z_2^{a_2})$$
$$=> (z_1^{a_1} \circledast z_2^{a_1}).(z_1^{a_2} \circledast z_2^{a_2}) \tag{2}$$

**Option 3:** In this, we select both pairwise inter-(pink, red) similarity

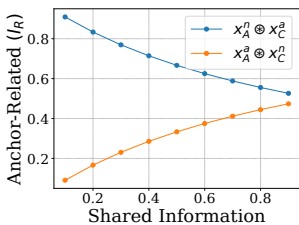

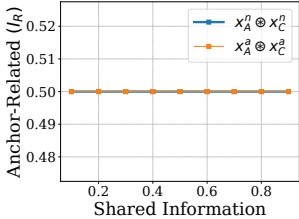

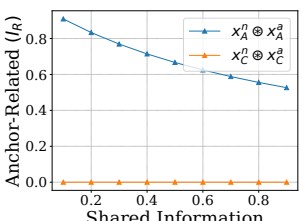

Figure 2: Option-1         Figure 3: Option-2         Figure 4: Option-3

Figure 5: MIMV options

$\left(z_1^{a_1}.z_2^{a_1}\right),\left(z_1^{a_2}.z_2^{a_2}\right),\left(z_1^{a_2}.z_2^{a_1}\right),\left(z_1^{a_1},z_2^{a_2}\right).$

$$=> \left(z_1^{a_1}.z_2^{a_1}\right) \circledast \left(z_1^{a_1}.z_2^{a_2}\right) \circledast \left(z_1^{a_2}.z_2^{a_1}\right) \circledast \left(z_1^{a_2},z_2^{a_2}\right)$$
$$=> z_1^{a_1}.\left(z_2^{a_1} \circledast z_2^{a_2}\right) \circledast z_1^{a_2}.\left(z_2^{a_1} \circledast z_2^{a_2}\right)$$
$$=> \left(z_1^{a_1} \circledast z_1^{a_2}\right).\left(z_2^{a_1} \circledast z_2^{a_2}\right) \tag{3}$$

The design of these possible combinations is given in Figure 22.

### A.3    SHARED INFORMATION ANALYSIS:

Let the information volume $V$ be the measurement of the anchor-related information contained within a representation. Volume $V = 1 \ (max)$, when the representation retains complete information as passed from its view, while $V = 0 \ (min)$, when there is complete loss of information. Other information volume lies between 0 and 1.$(min) \ 0 \leq \theta(z) \leq 1 \ (max)$. $\theta$ is measuring the volume and $z$ is an instance

$$\text{Information Ratio } (I_R(z)) = \frac{\text{representation volume}}{\text{total volume}} = \frac{\theta(z)}{V}$$

Using equation $(I(x;y) \geq I(z;y))$, we get $V(z_A^n) = 1$ , $V(z_C^n) = 1$. The reason that we are considering the normalized version of an image for both anchor and it's counterpart is that the normalized version of an image is invertible in nature, and therefore there is no loss of information. On the other hand, an augmented version of both images might lose information due to its non-invertible nature. To make things clear, $let \ V(z_A^a) = 0.8$ , $V(z_C^a) = 0.8$. Then the total information V for a pair of $(z_A, z_C)$ is measured by $V(z_A \circledast z_C) = V(z_A) + V(z_C)$. This analysis is needed to measure the shared information between pairs of representations as shown in Figure 5.

*Option-1:*

$$\left(z_A^n \circledast z_C^a\right),\left(z_A^a \circledast z_C^n\right)$$
$$I_R(A) = \frac{\theta(A)}{V} = \frac{\theta(A)}{V(z_A^n) + V(z_C^a)}, \frac{\theta(A)}{V(z_A^a) + V(z_C^n)}$$
$$I_R(A) = \frac{1}{1+0.8}, \frac{0.8}{0.8+1} = 0.55, 0.44$$

Both pairs have a volume of anchor-related information $0.55, 0.44$. A maximum shared information volume between our pairs can be $0.44$.

*Option 2:*

$$\left(z_A^n \circledast z_C^n\right).\left(z_A^a \circledast z_C^a\right)$$
$$I_R(A) = \frac{\theta(A)}{V} = \frac{\theta(A)}{V(z_A^n) + V(z_C^n)}, \frac{\theta(A)}{V(z_A^a) + V(z_C^a)}$$
$$I_R(A) = \frac{\theta(A)}{V} = \frac{1}{1+1}, \frac{0.8}{0.8+0.8} = 0.5, 0.5$$

Both pairs have a volume of anchor-related information $0.5, 0.5$. A maximum shared information volume between our pairs can be $0.5$.

*Option 3:*

$$(z_A^n \circledast z_A^a).(z_C^n \circledast z_C^a)$$

$$I_R(A) = \frac{\theta(A)}{V} = \frac{\theta(A)}{V(z_A^n) + V(z_A^a)}, \frac{\theta(A)}{V(z_C^n) + V(z_C^a)}$$

$$I_R(A) = \frac{\theta(A)}{V} = \frac{1.8}{1 + 0.8}, \frac{0}{1 + 0.8} = 1, 0$$

Both pairs have a volume of anchor-related information $1, 0$. A maximum shared information volume between our pairs can be $0$. This instance represents the information ratio kept by each pair with respect to the anchor image. As information loss decreases, the shared information increases in Figure 2 Figure 3 does not reflect any change in the information shared. Figure 4 has one bad representation that does not have information on the anchor. Using this analysis, we find that the representation pair in Figure 2 is the most promising pair to choose.

A.4  ELIMINATION BY OUR LOSS FUNCTION:

In order to give a detailed explanation for the MIMV loss 4, we provide additional information. In Figure 9, each image vividly demonstrates our loss in flow. In the First image 6, we have the similarities of pairs as illustrated in Figure 2, Option-1 (in the main section of the paper). The white circles are those that are considered for the loss calculation as they lie within the threshold while on the other hand, black circles lie outside the threshold and denotes those extreme cases which need not be considered. The vertical and horizontal lines represents the threshold for extreme cases. Later, in the second 7 and third figures 8, our loss function continuously eliminates the features that are not necessary for learning, providing a clear visual representation of the data flow.

$$\text{Let } S = Sim\left((z_A^n \circledast z_C^a).(z_A^a \circledast z_C^n)\right)$$

$$S(i,j) = \{S_{i,j}.\mathbb{1}_{[\lambda_l, \lambda_h]} S_{i,j}\}$$

$$\text{where } \mathbb{1}_{[\lambda_l, \lambda_h]} S_{i,j} = \begin{cases} \mathcal{S}_{i,j}, & \text{if } \lambda_l \leq S_{i,j} \leq \lambda_h \\ 0, & \text{otherwise} \end{cases}$$

$$\text{Final Loss: } \mathcal{L}_{i,j} = -log\left[\frac{e^{S/\tau}}{\sum_{k=1}^{2N} \mathbb{1}_{\{i \neq k\}} e^{S/\tau}}\right] \tag{4}$$

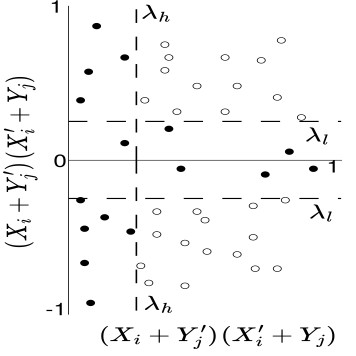

Figure 6: Flow-1

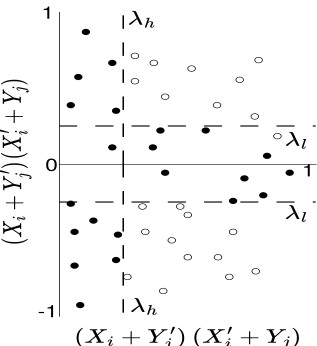

Figure 7: Flow-2

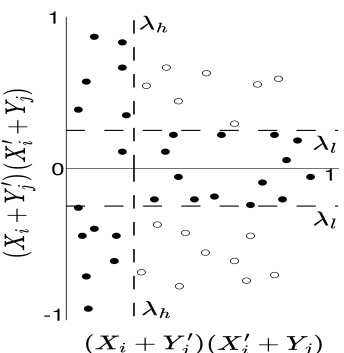

$(X_i + Y'_j)(X'_i + Y_j)$

Figure 8: Flow-3

Figure 9: Cosine Similarity of MIMV pairs. Figure 6 shows the initial cosine similarity of MIMV pairs, where $\lambda_h$ and $\lambda_l$ works as filter to cosider good examples for loss calculation. Later 7, and in 8 shows the upcoming iterations, in which more similarity pairs are filtered out by the loss function.

### A.4.1 LOSS THRESHOLD ANALYSIS ($\lambda$ ANALYSIS)

To find the best possible threshold, several experiments were carried out using the Cifar100-LT dataset using different combinations of $\lambda_h$ and $\lambda_l$ as can be seen from 10. The values used for $\lambda_h$ range from 1 to 0.7 and for $\lambda_l$ range from 0 to 0.4. We observe that a higher threshold (lower $\lambda_h$ and higher $\lambda_l$) results in unstable learning, while a lower threshold (higher $\lambda_h$ and lower $\lambda_l$)) does not improve performance. The values of 0.1 ($\lambda_l$) and 0.9 ($\lambda_h$) give the best result, as highlighted in 10. All experiments were carried out over 500 epochs. For large data sets such as ImagNet-LT, we observe that $\lambda_l(0.1)$ is more crucial than $\lambda_h(1.0)$ as $\lambda_h$ is more sensitive to numerical instability.

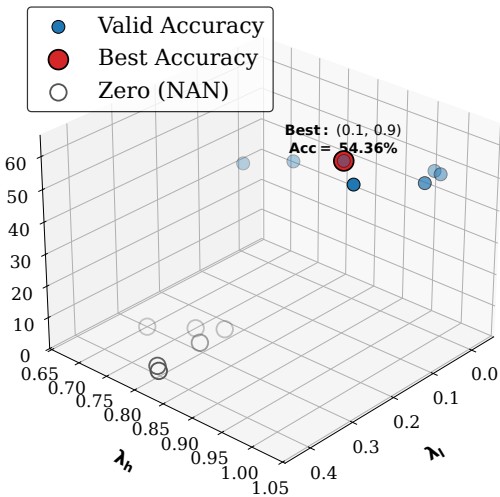

Figure 10: Threshold $\lambda_h$ and $\lambda_l$ analysis

### A.5 SIGNIFICANCE OF PROPOSED LOSS FUNCTION WITH MIMV FOR REPRESENTATION LEARNING

Figure 11 to Figure 14 show the advantage of the proposed loss function over the traditional NT-Xent loss. In the figures, MIMV denotes MIMV plus NT-Xent loss, while MIMV(updated) refers to MIMV plus the proposed loss. These experiments were carried out on Cifar100-LT for 2000 epochs. An imbalance factor of 0.01 with exponential distribution is used in Cifar100 to generate Cifar100-

LT. Cifar100-LT is divided into three parts, *Frequent, Medium, Few*. Frequent contains classes that have the highest number of images and is considered as head. Similarly, medium contains less number of images when compared to the head and few denotes the tail classes having the least number of images[1]. From the figures it can be observed that in all scenarios the proposed loss function exhibits superior behavior compared to the NT-Xent loss and performs best when it comes to tail classes (Figure 13).

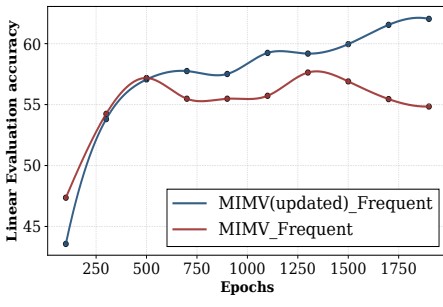

Figure 11: Frequent Classes

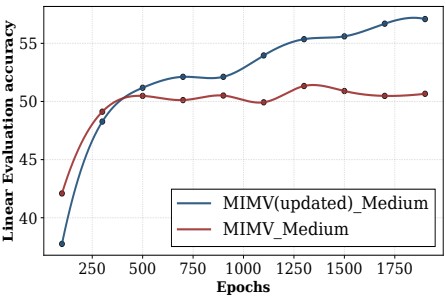

Figure 12: Medium Classes

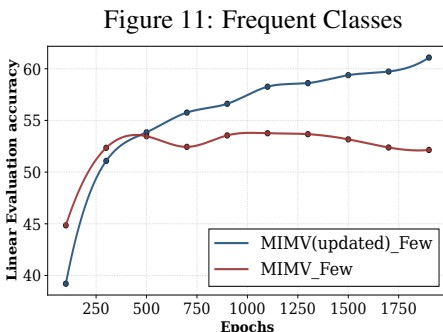

Figure 13: Few Classes

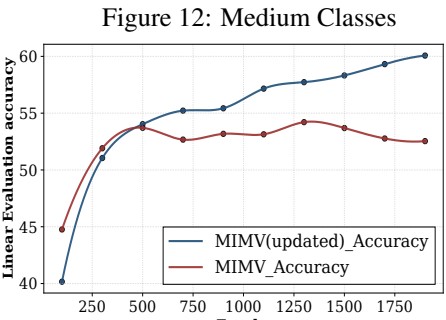

Figure 14: Linear Evaluation accuracy

## A.6 MUTUAL INFORMATION ANALYSIS

It is well known that in long-tail datasets mutual information (MI) is dominated by head classes because of their probability mass. In such cases, even with poor performance on the tail classes, high MI can be achieved. This means that high MI cannot be equated with high accuracy as far as long-tail data is concerned. The results outlined in Figures(Figure 15 to Figure 17) demonstrate this claim in which the proposed MIMV, MIMV + updated loss achieved lower MI, but this is not the case as can be seen from Figure 18 where high accuracy is achieved even with low MI.

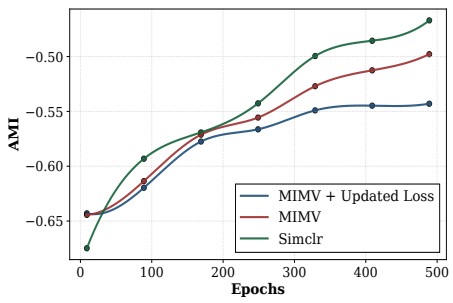

Figure 15: Adjusted Mutual Information

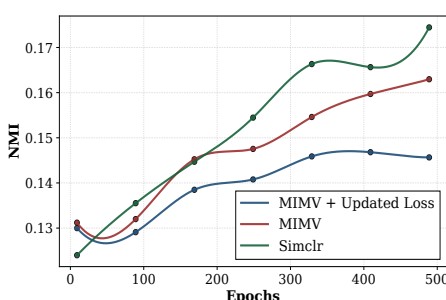

Figure 16: Normalized Mutual Information

---

[1]For imbalanced factor 0.01, generally frequent classes are the first 36 classes, medium is next 34 classes and few consist of the last 30 classes.

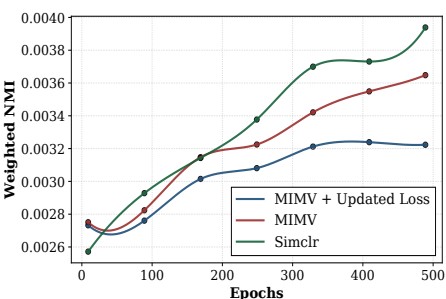

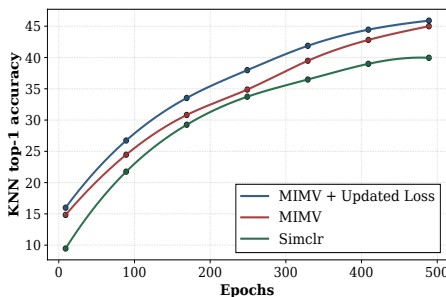

Figure 17: Weighted NMI

Figure 18: KNN Top-1 Accuracy on Cifar100

### A.7 ALIGNMENT AND UNIFORMITY ANALYSIS

In order to measure representation quality, two measures (alignment and uniformity) have been suggested by Tongzhou Wang & Philip Isola (Understanding Contrastive Representation Learning through Alignment and Uniformity on the Hypersphere, ICML-2020). The pattern of pulling positive examples closer (`alignment`) while pulling apart negatives that are irrelevant (`uniformity`) are two key properties that any contrastive loss function should satisfy. From Figure19 to Figure21 it can be seen that the proposed MIMV framework with updated loss satisfies both alignment and uniformity compared to other SSL frameworks.

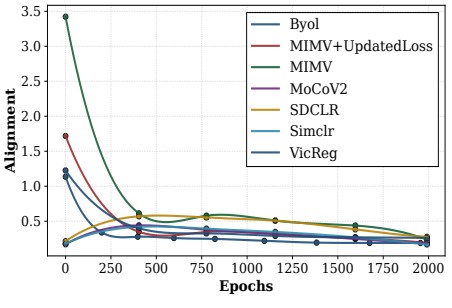

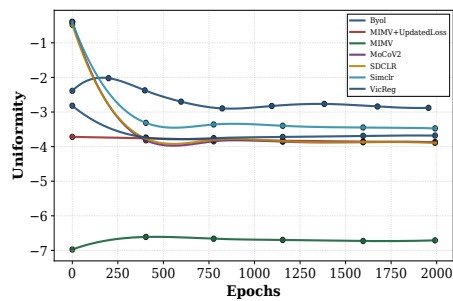

Figure 19: Alignment

Figure 20: Uniformity

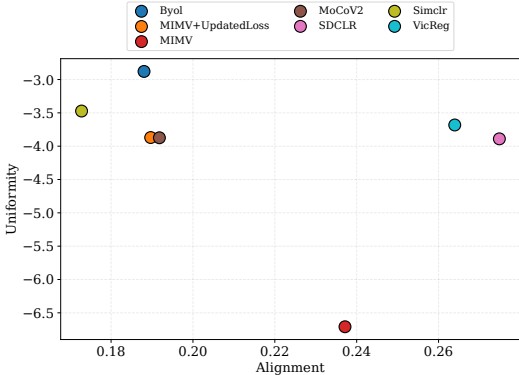

Figure 21: Alignment and Uniformity analysis

### A.8 T-SNE REPRESENTATIONS

Figures, (Figure22 to Figure28) show t-SNE representations of embeddings related to Cifar10-LT. We can visualize from these figures that the proposed framework(Figure22) has better discriminative representation while MIMV plus the proposed loss (Figure25) has even better discriminative ability.

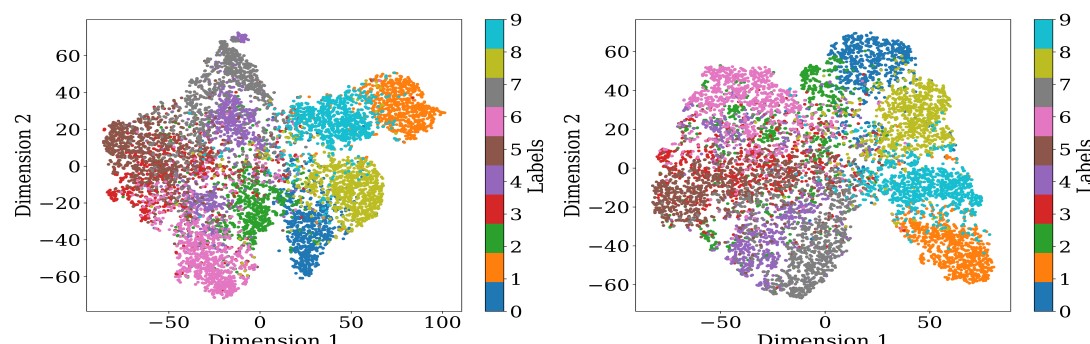

Figure 22: MIMV          Figure 23: BYOL

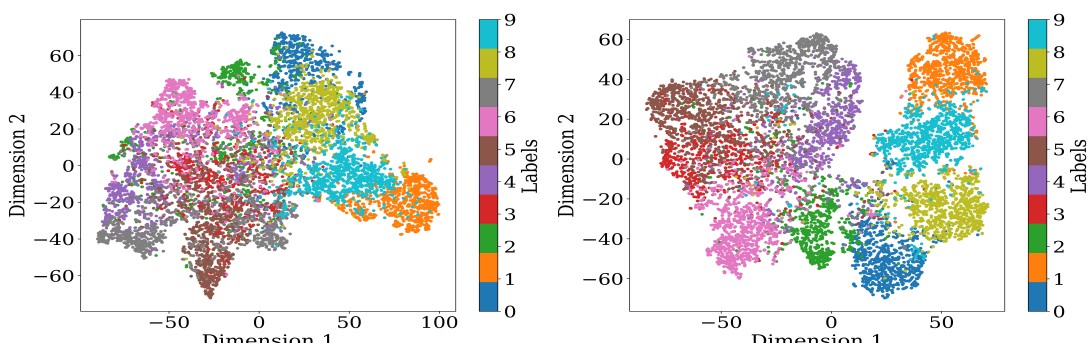

Figure 24: VicReg          Figure 25: MIMV + Updated Loss

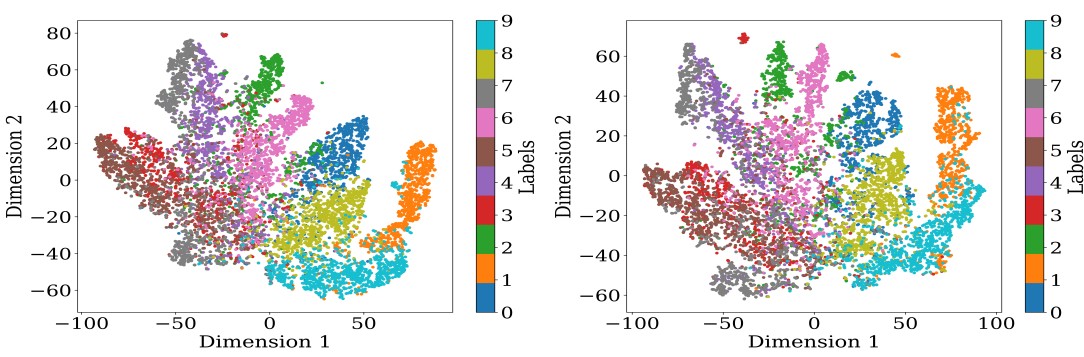

Figure 26: SDCLR          Figure 27: Simclr

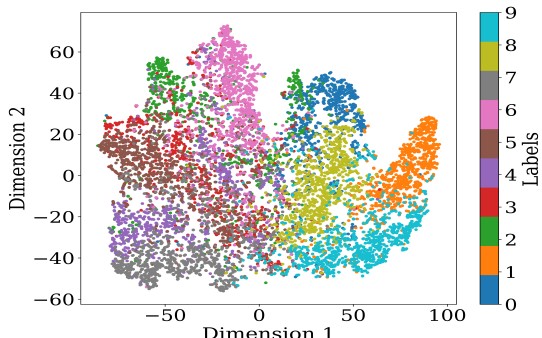

Figure 28: MoCoV2

### A.9 PER CLASS ACCURACY ANALYSIS ON CIFAR-10

The per class accuracy analysis on Cifar10-LT is given in Figure29. From the figure it can be observed that the proposed MIMV plus updated loss is consistently having better results over each class.

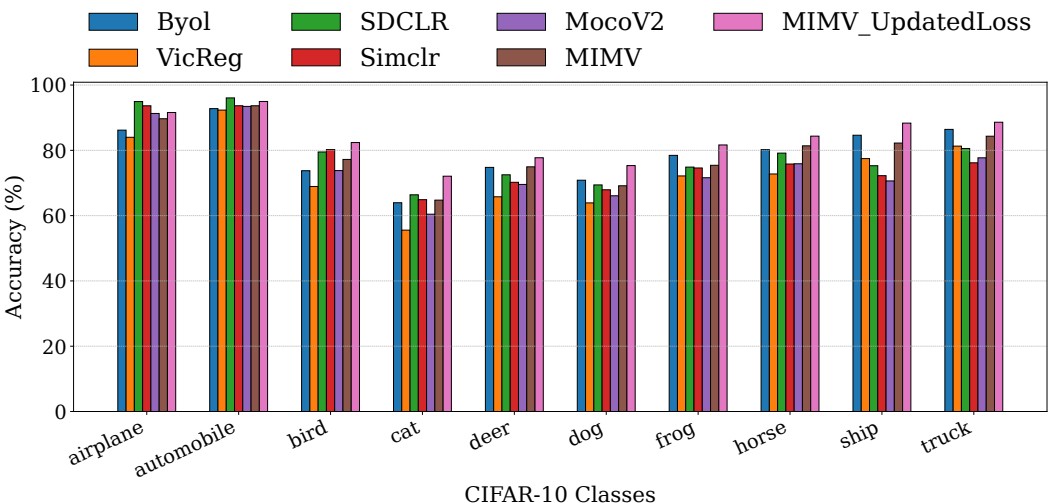

Figure 29: Caption

### A.10 MUTUAL INFORMATION WITH NT-XENT LOSS

In order to prove Mutual Information as given in Equation 5, we provide the supporting proof as follows: When we reexamine the NT-Xent environment, it can be seen that two positive pairs are chosen among $2N$ samples. This happens because NT-Xent averages the loss in both directions, such as $(z_i, z_j)$ and $(z_j, z_i)$.

$$\mathcal{L}_{i,j} = -\log \frac{\exp\left(\text{sim}(z_i, z_j)/\tau\right)}{\sum_{k=1}^{2N} \mathbb{1}_{[k \neq i]} \exp\left(\text{sim}(z_i, z_k)/\tau\right)} \tag{5}$$

Mutual information between $X$ and $Y$ is defined as:

$$I(X;Y) = \mathbb{E}_{(p_x, p_y)}\left[\log \frac{p(x,y)}{p(x)p(y)}\right] \tag{6}$$

This measures how much knowing $X$ reduces uncertainty about $Y$ and vice versa.

Let $x$ be an anchor image and our objective is to find which is the true positive out of $N$ samples.

**Formulating as a classification problem:**

Given an anchor image $x$, we want to predict which of the samples $N$ is the true positive $y$ (the one drawn from the joint $p(x, y)$). The other $2N - 2$ samples are drawn from the marginal $p(y)$ — i.e., they're negatives.

Let $(x, y_1) \sim p(x, y)$ be a true pair and let $y_2, \ldots, y_{2N-1} \sim p(y)$ be negatives, sampled independently of $x$. Let $y_{1:2N-1} = \{y_1, \ldots, y_{2N-1}\}$ be the complete set of candidates.

We define a classification task: Given $x$ and $\{y_1, \ldots, y_{2N-1}\}$, predict which is the positive.

This gives:

$$p(i = 1 | x, y_{1:2N-1}) = \frac{p(x, y_1)}{\sum_{k=1}^{2N-1} p(x) p(y_k)}$$

Since we do not know the exact distributions $p(x, y), p(x), p(y)$, we approximate this ratio with the Radon-Nikodym derivative:

$$f(x, y) \propto \frac{p(x, y)}{p(x) p(y)} \tag{7}$$

Our critic function in NT-Xent loss is defined as:

$$f(x, y) = \exp(\text{sim}(x, y)/\tau)$$

Using this, we can reshape our softmax classifier as follows:

$$q(i = 1 | x, y_{1:2N-1}) = \frac{f(x, y_1)}{\sum_{k=1}^{2N-1} f(x, y_k)}$$

NT-Xent loss is the negative log-likelihood of the correct class:

$$\mathcal{L}_{i,j} = -\mathbb{E}_{(x, y_1) \sim p(x,y), \{y_k\}_{k=1}^{2N-1} \sim p(y)} \left[ \log \frac{f(x, y_1)}{\sum_{k=1}^{2N-1} f(x, y_k)} \right] \tag{8}$$

Now, consider:

$$I(X; Y) = \mathbb{E}_{p^-} \left[ \log \frac{p(x, y)}{p(x) p(y_1)} \right]$$

We approximate the marginals using a Monte Carlo estimator.

$$p(x) p(y_1) \approx \frac{1}{2N - 1} \sum_{k=1}^{2N-1} p(x) p(y_k)$$

Thus:

$$\frac{p(x, y_1)}{p(x) p(y_1)} \approx \frac{p(x, y_1)}{\frac{1}{2N-1} \sum_{k=1}^{2N-1} p(x) p(y_k)}$$

Taking log:

$$\log \left[ \frac{p(x, y_1)}{p(x) p(y_1)} \right] = \log \left[ \frac{p(x, y_1)}{\frac{1}{2N-1} \sum_{k=1}^{2N-1} p(x) p(y_k)} \right]$$

$$= \log(2N - 1) + \log \left[ \frac{p(x, y_1)}{\sum_{k=1}^{2N-1} p(x) p(y_k)} \right]$$

By Jensen's inequality (since log is concave):

$$\log \mathbb{E}[Z] \geq \mathbb{E}[\log Z] \tag{9}$$

Apply this and substitute from equation 8:

$$\mathbb{E}_{p^-}\left[\log \frac{p(x, y_1)}{\frac{1}{2N-1}\sum_{k=1}^{2N-1} p(x)p(y_k)}\right] \geq \mathbb{E}_{p^-}\left[\log \frac{f(x, y_1)}{\frac{1}{2N-1}\sum_{k=1}^{2N-1} f(x, y_k)}\right]$$

$$I(X;Y) \geq \log(2N-1) + \mathbb{E}_{p^-}\left[\log \frac{f(x, y_1)}{\sum_{k=1}^{2N-1} f(x, y_k)}\right]$$

$$= \log(2N-1) - \mathcal{L}_{\text{NT-Xent}}$$

$$I(X;Y) = log(N) - \mathcal{L}_{NT_Xent} + c \tag{10}$$

### A.11 MUTUAL INFORMATION WITH MIMV LOSS

In the MIMV environment, we can see that two complementary representations $(z_A^n, z_C^a), (z_A^n, z_C^a)$ of normalized and augmented versions of two images are chosen as positive among the 2N samples. $(z_A^n, z_A^a)$ are the representations of $x_A^n, x_A^a \in x_A$(anchor) and $(z_A^n, z_A^a)$ are the representations of $x_C^n, x_C^a \in x_C$(counterpart).

$$\mathcal{L}_{i,j} = \frac{e^{sim((z_A^n, z_C^a), (z_A^a, z_C^n))/\tau}}{\sum_{k=1}^{2N} 1_{i \neq k}\, e^{sim((z_A^n, z_C^a), (z_A^a, z_C^n))/\tau}}$$

**Formulating a classification problem:** Given an anchor image $x_A$ and a counterpart image $x_C$ their representations can be given as $z_A^n, z_C^a \in x_A$ and $z_C^n, z_C^a \in x_C$. Later, they form a pair $(x_A^n \circledast y_C^a)$ of representations, and we want to predict which of the $N$ samples is the true positive $(z_A^n \circledast z_C^a)$ (the one drawn from the joint $p(z_A^n \circledast z_C^a, z_A^a \circledast z_C^n)$). The other $2N-2$ samples are drawn from the marginal $p(z_A^a \circledast z_C^n)$ — i.e., they are negatives.

Let $(z_A^n \circledast z_C^a, z_A^a \circledast z_C^n) \sim p(z_A^n \circledast z_C^a, z_A^a \circledast z_C^n)$ be a true pair and let $(z_A^a \circledast z_C^n)_2, \ldots, y_{2N-1} \sim p(z_A^a \circledast z_C^n)$ be negatives, sampled independently of $z_A^n \circledast z_C^a$. Let $(z_A^a \circledast z_C^n)_{1:2N-1} = \{(z_A^a \circledast z_C^n)_1, \ldots, (z_A^a \circledast z_C^n)_{2N-1}\}$ be the complete set of candidates.

We define a classification task: Given $z_A^n \circledast z_C^n$ and $\{(z_A^a \circledast z_C^n)_1, \ldots, y_{2N-1}\}$, predict which is the positive.

This gives:

$$p(i = 1|(z_A^n \circledast z_C^a), (z_A^a \circledast z_C^n)_{1:2N-1}) = \frac{p((z_A^n \circledast z_C^a), (z_A^a \circledast z_C^n)_1)}{\sum_{k=1}^{2N-1} p(z_A^n \circledast z_C^a)p(z_A^a \circledast z_C^n)_k}$$

Since we do not know the exact distributions $p((z_A^n \circledast z_C^a), (z_A^a \circledast z_C^n)), p((z_A^n \circledast z_C^a)), p((z_A^a \circledast z_C^n))$, we approximate this ratio with the Radon-Nikodym derivative:

$$f((z_A^n \circledast z_C^a), (z_A^a \circledast z_C^n)) \propto \frac{p((z_A^n \circledast z_C^a), (z_A^a \circledast z_C^n))}{p((z_A^n \circledast z_C^a))p((z_A^a \circledast z_C^n))}$$

Our critic function in MIMV loss is defined as:

$$f((z_A^n \circledast z_C^a), (z_A^a \circledast z_C^n)) = \exp(sim((z_A^n \circledast z_C^a), (z_A^a \circledast z_C^n))/\tau)$$

Using this, we can reshape our softmax classifier as follows:

$$q(i = 1|(z_A^n \circledast z_C^a), (z_A^a \circledast z_C^n)_{1:2N-1}) = \frac{f((z_A^n \circledast z_C^a), (z_A^a \circledast z_C^n)_1)}{\sum_{k=1}^{2N-1} f((z_A^n \circledast z_C^a), (z_A^a \circledast z_C^n)_k)}$$

Since MIMV loss is the negative log-likelihood of the correct class:

$$\mathcal{L}_{i,j} = -\mathbb{E}_{((z_A^n \circledast z_C^a), z_C^n) \sim p((z_A^n \circledast z_C^a),(z_A^a \circledast z_C^n)), \{(z_A^a \circledast z_C^n)_k\}_{k=1}^{2N-1} \sim p((z_A^a \circledast z_C^n))} \left[ \log \frac{f((z_A^n \circledast z_C^a), (z_A^a \circledast z_C^n)_1)}{\sum_{k=1}^{2N-1} f((z_A^n \circledast z_C^a), (z_A^a \circledast z_C^n)_k)} \right]$$

$$(11)$$

Since the classification formulation reveals that the negative log-liklelihood of MIMV loss 11 is an instance of the negative log-likelihood of the NT-Xent loss 8, its mutual information lower bound is derived as follows:

$$I((z_A^n \circledast z_C^a); (z_A^a \circledast z_C^n)) \geq \log(2N - 1) - \mathcal{L}_{\text{MIMV}}$$
$$I((z_A^n \circledast z_C^a), (z_A^a \circledast z_C^n)) \geq log(N) - \mathcal{L}_{MIMV} + c \qquad (12)$$

Our code is available at https://github.com/anonymouspersoncscv/MTTV.git.