# OpenReview forum: "Maximally Useful and Minimally Redundant: The Key to Self Supervised Learning for Imbalanced Data"
_ICLR.cc/2026/Conference — Submitted to ICLR 2026_

### Official Review · Reviewer_VRHb · 2025-10-25

**Soundness:** 2
**Presentation:** 2
**Contribution:** 2
**Rating:** 4
**Confidence:** 2

**Summary:**

In contrastive self-supervised learning (CSSL), when the data follows a long-tailed or imbalanced distribution, the traditional approach of generating two augmented views from the same image performs poorly on tail classes (categories with few samples). To enable self-supervised multi-view frameworks to extend to settings involving more than two views (MTTV), this paper proposes a method to filter out extreme or uninformative features, thereby facilitating the learning of better representations. Experiments on imbalanced self-supervised datasets demonstrate that the proposed method achieves strong performance.

**Strengths:**

S1: The motivation of this work is reasonable. Extending from two-view contrastive learning to cross-image and cross-view fusion is intuitive and practically meaningful.

S2: From an information-theoretic perspective, exploring representations that are maximally useful and minimally redundant is a logical and well-grounded research direction.

S3: The experimental results demonstrate the effectiveness of the proposed method. Across several imbalanced datasets (CIFAR10-LT, CIFAR100-LT, and ImageNet-LT (1K)), the method consistently achieves performance improvements over existing methods.

**Weaknesses:**

W1: The theoretical analysis relies on a coarse approximation where the information content of different views is simply summed. This makes Section 3.1 appear somewhat superficial. A more rigorous treatment is needed, including a formal derivation or conditions under which this approximation holds.

W2: The writing of the paper is somewhat verbose, which obscures the main contributions. In particular, a substantial portion of the paper (starting around line 063) is devoted to reintroducing methods such as SimCLR, which appears redundant.

W3: As a consequence of (2), the authors do not clearly isolate and evaluate the contribution of each key component. The experimental section therefore lacks clarity and persuasive power, and would benefit from more systematic ablation studies to demonstrate the necessity of each design choice.

**Questions:**

Given that the Related Work section is not very detailed, could the authors supplement it with a brief summary of recent papers on long-tailed self-supervised and semi-supervised learning, and select appropriate strong baseline algorithms to add to the experimental section?

---

> ### Author Response · Authors · 2025-11-20
>
> ### The theoretical analysis relies on a coarse approximation where the information content of different views is simply summed. This makes Section 3.1 appear somewhat superficial. A more rigorous treatment is needed, including a formal derivation or conditions under which this approximation holds.
>
> **Reply:** We would like to draw the attention of the reviewer to Section-1. In Figure-2, first we highlight Simclr with two augmented views which is how the  standard Simclr works (denoted as Simclr in the figure). Thereafter results related to Simclr + Normalized + Augmented views (referred to as Simclr + NA in the figure). After that MIMV (earlier denoted as MTTV) with both augmented views is shown(MIMV + AA). In the end, we demonstrate MIMV with normalized and augmented views (referred to as MIMV + NA). These results include NT-Xcent loss without threshold.
>
> Model | Linear Evaluation Top-1 Accuracy
>
> Simclr | 44.49
>
> Simclr + NA | 47.31
>
> MIMV + AA | 48.01
>
> MIMV + NA | 52.84
>
> MIMV + NA + Threshold Loss | 60.18
>
> ### The writing of the paper is somewhat verbose, which obscures the main contributions. In particular, a substantial portion of the paper (starting around line 063) is devoted to reintroducing methods such as SimCLR, which appears redundant.
>
> **Reply:** We have rewritten the Abstract, Introduction as well as the Related work sections.
>
> ### As a consequence of (2), the authors do not clearly isolate and evaluate the contribution of each key component. The experimental section therefore lacks clarity and persuasive power, and would benefit from more systematic ablation studies to demonstrate the necessity of each design choice.
>
> **Reply:** We have rewritten the introduction as well as the Related work Section. More ablation studies have been carried out and all these changes will be reflected in the updated version.
>
> ### Given that the Related Work section is not very detailed, could the authors supplement it with a brief summary of recent papers on long-tailed self-supervised and semi-supervised learning, and select appropriate strong baseline algorithms to add to the experimental section?
>
> **Reply:** Yes definitely.
>
> We have updated the experimental section with the following results based on ImageNet-LT
>
> Byol-2020 | 43.16 %
>
> VicReg-2022 | 38.24\%
>
> SDCLR-2021 | 46.62\%
>
> SIMCLR-2020 | 46.81\%
>
> MoCoV2-2020 | 49.50\%
>
> Ours (MIMV)-2025 | 52.90\%
>
> We have updated the experimental section with the following results based on CIFAR-100-LT
>
> SIMCLR-2020  | 44.85\%
>
> MoCoV2-2020 | 46.37\%
>
> Byol-2020 | 47\%
>
> VicReg-2022 | 45.26\%
>
> SWaV-2020 | 48.86\%
>
> BCL-I-2022 | 52.22\%
>
> SDCLR-2021 | 54.94\%
>
> FASSL-2023 | 55.27\%
>
> Ours(MIMV-Concat)-2025 | 58.89\%
>
> Ours(MIMV(Sum)-2025 | 60.18\%
>
> We have updated the experimental section with the following results based on CIFAR-10-LT
>
> SIMCLR-2020 | 76.77\%
>
> MoCoV2-2020 | 74.76\%
>
> Byol-2020 | 75.66\%
>
> VicReg-2022 | 73.32\%
>
> SWaV-2020 | 76.60\%
>
> Sim-SIam - 2022 | 81.40\%
>
> SDCLR-2021 | 80.49\%
>
> FASSL-2023 | 80.69\%
>
> Ours(MIMV(Sum)-2025 | 83.66\%
>
> Long-tail learning in self-supervised frameworks is under-explored. However there are some recent training techniques available to facilitate long-tail learning in SSL, for instance,  Kukleva et al., ICLR, 2023 and Bai et al., ICLR, 2023. The main contribution of Kukleva et.al, is with respect to the introduction of a dynamic/class frequency-aware temperature parameter for NT-Xent/InfoNce that can be fitted within a SIMCLR/MOCO kind of framework and as such do not propose a novel framework as MIMV (earlier MTTV). Bai et al., also do not design a new contrastive model but only propose a novel training strategy that can be fitted into an existing framework like SIMCLR/SDCLR. Incorporating these techniques in our framework could have resulted in better performance but the main motivation was to come up with a new CSSL  framework that can accommodate multi image multi view assumption.

---

> > ### Comment · Reviewer_VRHb · 2025-11-26
> >
> > Thank you for your response. I believe the current version still has issues in the writing and experimental sections, so I maintain my score.

---

### Official Review · Reviewer_FVwq · 2025-10-27

**Soundness:** 2
**Presentation:** 1
**Contribution:** 2
**Rating:** 2
**Confidence:** 3

**Summary:**

The paper proposes MTTV, a self-supervised contrastive framework for long-tailed data that: (i) forms compact fused representations from more-than-two views (augmented and normalized), and (ii) masks extreme similarities via a thresholded InfoNCE variant. Experiments on CIFAR10-LT/100-LT and ImageNet-LT subsamples report linear-evaluation gains over prior SSL baselines such as SimCLR, BYOL, SwAV, SDCLR, and FASSL.

**Strengths:**

* Shows clear empirical gains on CIFAR-LT and ImageNet-LT over several standard SSL baselines.
* The proposed “extreme-feature elimination” (thresholded similarity filtering) is a simple idea that appears to stabilize training and may reduce representation collapse.

**Weaknesses:**

* Missing Comparison to Multi-View Self-Supervised Learning: The manuscript repeatedly claims novelty in using more-than-two views and redundancy suppression, yet fails to compare to the extensive multi-view SSL literature where these principles were introduced and validated, including:
    * An Embedding-Dynamic Approach to Self-Supervised Learning (WACV 2023)
    * VICReg (ICLR 2022)
    * SwAV (NeurIPS 2020)
    * MIL-NCE: End-to-End Learning of Visual Representations from Uncurated Instructional Videos (CVPR 2020)
    * Learning by Sorting: Self-Supervised Learning with Group Ordering Constraints (ICCV 2023) Moreover, the proposed “feature filtering” closely resembles hard-negative mining; a direct comparison to such methods is missing.
* Incorrect citation: FASSL is listed as an ICCV proceedings paper; it is actually an ICCV 2023 Workshop  paper. Please correct this.
* Missing baselines and discussion: The paper omits “Contrastive Learning with Boosted Memorization” (ICML 2022), which is conceptually close. It should be included in comparisons and discussed in detail. Additionally, evaluations should consider more recent visual-only backbones such as DINOv3 and SigLIP-v2 to ensure fair benchmarking.
* Motivation vs. Method Mixing: The introduction reads like a methods section, filled with equations and implementation details but little conceptual framing or problem motivation. The broader context of long-tailed SSL is not well developed.
* Questionable novelty framing: The abstract credits a LeCun comment as inspiration for multi-view design, while multi-view SSL has been explored for years (see above). This framing misrepresents the originality of the idea.
* Figures and clarity: Figures are small, crowded with equations, and not self-contained. They do not effectively illustrate the pipeline or contributions.
* Insufficient ablations: No analysis disentangles the impact of the fusion mechanism, thresholding, or other design choices; thus, the source of reported gains remains unclear.

**Questions:**

see above

---

> ### Author Response · Authors · 2025-11-20
>
> ### Missing Comparison to Multi-View Self-Supervised Learning: The manuscript repeatedly claims novelty in using more-than-two views and redundancy suppression, yet fails to compare to the extensive multi-view SSL literature where these principles were introduced and validated, including:
>
> An Embedding-Dynamic Approach to Self-Supervised Learning (WACV 2023)
>
> VICReg (ICLR 2022)
>
> SwAV (NeurIPS 2020)
>
> MIL-NCE: End-to-End Learning of Visual Representations from Uncurated Instructional Videos (CVPR 2020)
>
> Learning by Sorting: Self-Supervised Learning with Group Ordering Constraints (ICCV 2023). Moreover, the proposed "feature filtering" closely resembles hard-negative mining; a direct comparison to such methods is missing.
>
> **Reply:** Experiments are being carried out for comparison of our proposed model with VicReg, SwAV, WACV-2023 as well as the learning by sorting paper. We feel that MIL-NCE is out of the scope for this work.
>
> For Cifar-10
>
> VicReg | 73.32\%
>
> SwaV  | 76.60 \%
>
> For Cifar-100
>
> VicReg | 45.26\%
>
> SwaV  | 48.86 \%
>
> Learning by sorting and Embedding Dynamic Approach to SSL (WACV-23) could not be implemented.
>
> ### Incorrect citation: FASSL is listed as an ICCV proceedings paper; it is actually an ICCV 2023 Workshop paper. Please correct this.
>
> **Reply:** Citation corrected.
>
> ### Missing baselines and discussion: The paper omits “Contrastive Learning with Boosted Memorization” (ICML 2022), which is conceptually close. It should be included in comparisons and discussed in detail. Additionally, evaluations should consider more recent visual-only backbones such as DINOv3 and SigLIP-v2 to ensure fair benchmarking.}
> **Reply:** Results related to "Contrastive Learning with Boosted Memorization" are already included in the paper in Section-4 (Table-2). DINOv3 has VIT as the backbone, whereas our primary focus is on Resnet-18 and Resnet-50 as the backbone. Compared to Resnet, VIT backbones are heavier in nature, and a comparison between the two is out of scope as far as this work is concerned. To our knowledge, SigLIP-v2 is a vision-language contrastive model and may not contribute to solving problems related to only images as in our work.
>
> ### Motivation vs. Method Mixing: The introduction reads like a methods section, filled with equations and implementation details but little conceptual framing or problem motivation. The broader context of long-tailed SSL is not well developed.
>
> **Reply:** We have rewritten the introduction and the same is incorporated in the updated version.
>
> ### Questionable novelty framing: The abstract credits a LeCun comment as inspiration for multi-view design, while multi-view SSL has been explored for years (see above). This framing misrepresents the originality of the idea.
>
> **Reply:** We are not attributing credit to Yan LeCun for Multi-view learning but for putting forward the idea of "More than Two Views" as part of the future work in one of his recent papers: Ravid Schwartz and Yann LeCun ("To Compress or Not to Compress-Self-Supervised Learning and Information Theory: A Review") which appeared in the journal "Entropy, volume 26, Number 252, 2024". The concept of MTTV (renamed as MIMV) is our own, and only the intuition is borrowed from Ravid Schwartz & Yan LeCun. In order to avoid confusion we have renamed MTTV to MIMV-Multi Image Multi View.
>
> ### Figures and clarity: Figures are small, crowded with equations, and not self-contained. They do not effectively illustrate the pipeline or contributions.
>
> **Reply:** We have improved the quality of the figures and the same has been incorporated in the updated version.
>
> ### Insufficient ablations: No analysis disentangles the impact of the fusion mechanism, thresholding, or other design choices; thus, the source of reported gains remains unclear.
>
> **Reply:** The ablation study related to filtering thresholds is given in the Appendix where $\lambda_{h}$ as higher limit and $\lambda_{l}$ as lower limit. The target is to select all similarity pairs for loss calculation and discard extreme feature similarity pairs.
>
> We carried out this analysis for CIFAR-100-LT. Similarly, ablation studies related to fusion type as well as per class behavior is carried out and given in the Appendix.

---

### Official Review · Reviewer_yEsw · 2025-10-30

**Soundness:** 1
**Presentation:** 1
**Contribution:** 1
**Rating:** 2
**Confidence:** 5

**Summary:**

The paper targets the robustness of self-supervised contrastive learning (CSSL) on long-tailed/class-imbalanced image data. It proposes “More Than Two Views (MTTV)”: unlike conventional CSSL that generates two augmented views of the same image for contrastive learning, the authors advocate using multi-views from two different images, mixing “invertible normalized views” and “non-invertible augmented views,” building more compact representations via “fusion” (sum or concat), and optimizing with a modified NT-Xent (InfoNCE)-style loss. Notably, this appears to be a very rushed submission; the presentation is very poor, making the paper extremely hard to read.

**Strengths:**

The authors completed a paper.

**Weaknesses:**

1. Setting aside the method itself, the presentation is very poor and falls far short of ICLR standards.  **I consider such a carelessly prepared submission disrespectful to reviewers.**

a. The abstract is filled with redundant, uninformative content. The core “what the paper does” is reduced to a single short sentence, which does not meet the norms of an abstract.

b. The introduction is excessively long (three pages). For example, there is no need to describe how contrastive learning works in such detail in the introduction; it gives the impression of padding pages.

c. Numerous formatting errors: an obvious extra space at line 59; equations are sometimes referenced as “Equation: 1” and sometimes as “Equation (1)”; “Knn” and “KNN” are mixed at line 432; and many other nonstandard expressions, such as missing punctuation after display equations. The authors need careful proofreading and revision.

d. The notation is very confusing, which makes the paper hard to read.

2. The assumption about the invertibility of the “normalized view” is vague: there is no clear specification of the transformation (pixel normalization or representation L2-normalization?), and DPI/MI are misused/oversimplified; the numerical analysis of “information volume/shared information” is subjectively assigned (e.g., 0.8) and lacks rigorous derivation or estimation methodology.

3. The paper treats fused representations from two different images as positive pairs without any mechanism to ensure semantic sharing, which contradicts the standard SSL assumption that positives are different views of the same instance, and is likely to cause optimization confusion or collapse.

4. Critical implementation details are missing: data augmentation pipeline, the exact definition and implementation of “normalized,” temperature, projection dimension, whether a queue/memory bank is used, composition of negatives, and how thresholding is applied to the InfoNCE denominator.

5. The paper does not explain why the proposed method should work specifically in long-tailed (LT) scenarios.

**Questions:**

None.

---

> ### Author Response · Authors · 2025-11-20
>
> ### Weaknesses:
> Setting aside the method itself, the presentation is very poor and falls far short of ICLR standards. I consider such a carelessly prepared submission disrespectful to reviewers.
>
> **Reply:** We have rewritten the paper.
> ### The abstract is filled with redundant, uninformative content. The core “what the paper does” is reduced to a single short sentence, which does not meet the norms of an abstract.
> **Reply:** The abstract has been rewritten:
>  ### The introduction is excessively long (three pages).
> **Reply:** Introduction has been updated (One page).
> ### Numerous formatting errors:
> **Reply:** All formatting errors is taken care in the updated version.
> ### The assumption about the invertibility of the “normalized view” is vague:
> **Reply:** We use the z-score normalization, which uses mean and standard deviation of the dataset. As normalization is invertible in nature, it tends not to lose any information of the respective image. Traditional SSL frameworks uses two augmented versions of a single image and therefore loss of information is bound to happen. Our augmentation strategy of having one normalized and one augmented view overcomes this loss. The superiority of this strategy is given in the paper in Section-2, Figure-2 as Simclr, Simclr+NA(normalised augmented).
> ### The paper treats fused representations from two different images as positive pairs without any mechanism to ensure semantic sharing, which contradicts the standard SSL assumption that positives are different views of the same instance, and is likely to cause optimization confusion or collapse.
> **Reply:** The first point is that the proposed model is setup in the backdrop of a multi-image, multi-view scenario (MIMV) and hence is different from the traditional multi-view setup. Initially, we start with two images from which two compact representations are generated which we denote as $z_{1}$ and $z_{2}$. One of these representations (for instance, $z_{1}$) shows dominance towards one image whereas the other representation (for instance, $z_{2}$) is less dominant towards the same image and vice-versa. So in our case positives are different representations (more dominant and less dominant pair) of the same image. How we arrived at this intuition of pair formation from the NT-Xent latent space is explained in Figure-1 (Section-1) and Figure-4 (Section-3).
> ### Critical implementation details are missing:
> **Reply:**
> To underline the precision of our approach, we present and validate our results on the ImageNet dataset. We used Resent18 and Resnet50 as our backbone architectures. We employed a three-layer MLP consisting of three fully connected layers with three batch normalization units and two RELU activations for the projection head. We used a batch size of $1024$ for Cifar10-LT and Cifar100-LT, while a batch size of $256$ was used for Imagenet-LT.
> We employ stochastic gradient descent with a learning rate of $3.0$ for Cifar10-LT and Cifar100-LT, while $0.5$ for Imagenet-LT subsamples made by sampling factor $s=0.125,0.25,0.50$ and $0.1$ for remaining subsamples $s=0.75,1.0$. We used a cosine decay learning rate with a warming of $10\%$ of the max epochs with a weight decay of $1e-4$ to learn a more robust representation. To update the weights of the target network ($encoder_k$), we used a momentum of $0.9$. All metrics reported here are trained with $300$ epochs in Imagenet-LT and $2000$ epochs in Cifar10-LT and Cifar100-LT, and an average of three runs. To eliminate extreme features by the limit of $(low)\;\lambda_l , \lambda_h\;(high)$, we use $\lambda_l = 0.1\; and\;\lambda_h = 0.9$ for Cifar10-LT and Cifar100-LT. For Imagenet-LT and their subsets, we use $\lambda_l = 0.1\; and\; \lambda_h =1.0$. We observed that we must carefully use this limit as it may result in collapse. We noticed that $\lambda_h$ is more sensitive to collapse.
> \textbf{All these details are given in Section-4 in the paper.}
> ### The paper does not explain why the proposed method should work specifically in long-tailed (LT) scenarios.
> **Reply:** Figure-2 reflects the implementation results related to CIFAR-100 LT dataset using Simclr plus our proposed augmentation strategy. There is a clear improvement of 2.8\% as far as Simclr accuracy is concerned. On further investigation, using our MIMV(earlier MTTV) design and augmentation strategy we could further improve the results. Since we know that the early dominance of majority classes is the main hindrance with regard to imbalanced datasets we devised strategies to eliminate extreme features which includes removing bad negatives and bad positives. Figure-6 clearly shows how this elimination of extreme features are carried out.

---

> ### Comment · Reviewer_yEsw · 2025-11-27
>
> Thank you for your response. I checked the recent PDF, but it did not address the issue I raised, so I decided to keep my rating.

---

### Official Review · Reviewer_P4oE · 2025-10-31

**Soundness:** 2
**Presentation:** 1
**Contribution:** 2
**Rating:** 2
**Confidence:** 4

**Summary:**

This paper addresses self-supervised learning (SSL) under imbalanced or long-tailed data distributions.
The authors argue that conventional contrastive SSL methods (e.g., SimCLR, MoCo) are biased toward head classes because they only form intra-image positive pairs.
To mitigate this, the paper introduces a “More-Than-Two-View” (MTTV) framework, inspired (as claimed) by Yann LeCun’s discussions on multi-view SSL.
The key idea is to construct cross-image feature fusion pairs between normalized and augmented views from different images, aiming to learn maximally useful and minimally redundant representations. The approach is evaluated on CIFAR-10/100-LT and ImageNet-LT.

**Strengths:**

1. Problem Motivation:  The paper focuses on an underexplored but important issue—class imbalance in self-supervised representation learning.


2. High-Level Idea:  Extending contrastive learning beyond single-image pairs to multi-image fusion introduces an interesting direction that may encourage feature diversity.

**Weaknesses:**

**1. Conceptual Vagueness:**
 The MTTV concept (“More Than Two Views”) is poorly defined, with no formal reference or clear operational meaning.
 The citation to LeCun is not substantiated by a verifiable publication.


**2. Weak Theoretical Grounding:**
 Although framed as information-theoretic, the paper does not compute or approximate mutual information. The Information Bottleneck formulation serves more as a conceptual narrative than a measurable objective.


**3. Lack of Experimental Rigor:**
- Section 4 Results section only reports main benchmark results (Tables 1–3).
- No ablation studies are provided to isolate the contribution of each component (fusion type, filtering thresholds, etc.).
- No analysis explaining why the method works (e.g., embedding visualization, MI analysis, or per-class behavior).


**4. Unconvincing Baseline Selection:**
 The comparisons are only made against standard SSL baselines (SimCLR, MoCo v2, BYOL, DINO) that do not address imbalance.
 Recent imbalance-aware SSL methods (e.g., Kukleva et al., ICLR, 2023, Bai et al., ICLR, 2023) are omitted.
 Consequently, the reported improvements (~2–5%) may reflect baseline selection bias.


**5. Shallow Related Work:**
 The Related Work section only discusses “imbalanced SSL” papers, ignoring the rich literature on long-tailed recognition [1, 2, 3] and SSL methods (SimCLR, MoCo). Including a broader set of prior works would make the paper more self-contained and better positioned within the field.

For example, some representative methods are:

[1] Learning Imbalanced Datasets with Label-Distribution-Aware Margin Loss, NeurIPS, 2019 --> Reweighting method

[2] Parametric Contrastive Learning, ICCV, 2021 --> Supervised Contrastive learning method

[3] The Majority Can Help The Minority: Context-rich Minority Oversampling for Long-tailed Classification, CVPR, 2022 --> Imbalance-aware augmentation method



**6. Speculative Causal Claims:**
 The link between “redundancy reduction” and “imbalance robustness” is asserted but not validated.

**Questions:**

Please refer to the weaknesses above.

1  How exactly is “More Than Two Views” defined, and can you provide a verifiable reference for LeCun’s supposed remark?

2. Have you quantitatively measured mutual information to support your “maximally useful” claim?

3. Ablations: What is the sensitivity of performance to the filtering thresholds (λₗ, λₕ)?  Is the fusion operation (sum vs. concat) critical to performance, and what are the trade-offs?

4. How does your approach compare against imbalance-aware SSL methods  (e.g., Kukleva et al., ICLR, 2023, Bai et al., ICLR, 2023) ?

---

> ### Author Response · Authors · 2025-11-20
>
> ### Conceptual Vagueness: The MTTV concept (“More Than Two Views”) is poorly defined, with no formal reference or clear operational meaning. The citation to LeCun is not substantiated by a verifiable publication
> **Reply:** We are extremely sorry for omitting out the reference related to Yan LeCun's paper. The concept of MTTV (renamed into MIMV - Multi Image Multi view in the updated version) is our own, the inspiration of which was derived from a paper written by Ravid Shwartz and Yann LeCun (To Compress or Not to Compress - Self Supervised Learning and Information Theory: A Review) which appeared in the journal \emph{Entropy, volume 26, Number 252, 2024}. Section 1 provides insights into the origins of the MIMV framework, and Section 3 contains the analysis and objectives related to the MIMV framework.
> ### Weak Theoretical Grounding:
> **Reply:** Since space was limited, the entire theoretical framework related to our work was uploaded as an Appendix, in which all derivations concerning approximating mutual information are shown.
> ### Lack of Experimental Rigor
> **Reply:** The ablation study related to filtering thresholds are given in the updated version of the Appendix. The target is to select all similarity pairs for loss calculation and discard extreme feature similarity pairs. We carried out this analysis for CIFAR-100-LT. Similarly, ablation studies related to fusion type as well as per class behavior are being carried out and . figures related to MI analysis are provided in the updated version in the Appendix.
> ### Unconvincing Baseline Selection:
> **Reply:** In the related work section, we have mentioned both Kukleva et al., ICLR, 2023 and Bai et al., ICLR, 2023. The main contribution of Kukleva et.al, is with respect to the introduction of a dynamic/class frequency-aware temperature parameter for NT-Xent/InfoNce that can be fitted within a SIMCLR/MOCO kind of framework and as such do not propose a novel framework as MIMV(earlier MTTV). Bai et al., also do not design a new contrastive model but only propose a novel training strategy that can be fitted into an existing framework like SIMCLR/SDCLR. In our case, we propose a novel CSSL framework in which these techniques can be accommodated.
> ### Shallow Related Work
> **Reply:** The related section is revised and is  included in the updated version of the main paper. Our intention is not to compare the results of supervised versus self-supervised for imbalanced data but to restrict ourselves to the CSSL family. The NeurIPS-2019 paper, ICCV-2021 as well as CVPR-2022 as pointed out by the reviewer falls under the category of supervised approach for imbalanced data but our focus is only on self supervised frameworks. We have removed the supervised baseline from all Tables in order to avoid that confusion and have added  experimental results  based on self supervised frameworks like SDCLR, SWAV, BYOL, SIMCLR and VicReg which are outlined in the updated version and a few of them are listed below:
>
> Byol 47\%
>
> VicReg 45.26\%
>
> SDCLR 54.94\%
>
> BCL-i 52.22 \%
>
> ### Questions:
>
> ### How exactly is “More Than Two Views” defined, and can you provide a verifiable reference for LeCun’s supposed remark?
>
> **Reply:** We have already answered this question in the explanation given above.
>
> ### Have you quantitatively measured mutual information to support your “maximally useful” claim?
> **Reply:**
> As shown in the figures given in the Appendix, measuring mutual information in the case of imbalanced data is not appropriate as compared to that of balanced data. The reason for this is that mutual information only measures view consistency and not class representation quality. MI measures how two augmented views are similar whereas long-tail problems need class aware representation quality (rare classes) which MI does not reflectc(BCL, Kukelava). In order to quantify representation quality in a self supervised manner, we have alignment and uniformity as a measure (Wangc\& Isola).
> ### Ablations: What is the sensitivity of performance to the filtering thresholds $(\lambda_{h}, \lambda){l})$? Is the fusion operation (sum/concat) critical to performance, and what are the trade-offs?
> **Reply:** The sensitivity results are given in the Appendix.
> ### How does your approach compare against imbalance-aware SSL methods (e.g., Kukleva et al., ICLR, 2023, Bai et al., ICLR, 2023) ?
> **Reply:** In the related work section, we had made reference to both Kukleva et al., ICLR, 2023 and Bai et al., ICLR, 2023. The main contribution of Kukleva et.al, is with respect to the introduction of a dynamic/class frequency-aware temperature parameter for NT-Xent/InfoNce that can be fitted within a SIMCLR/MOCO kind of framework and as such do not propose a novel framework as MIMV (earlier MTTV). Bai et al., also do not design a new contrastive model but only propose a novel training strategy that can be fitted into an existing framework like SIMCLR/SDCLR. We propose a novel architecture and a loss function.

---

### Official Review · Reviewer_8gZQ · 2025-11-01

**Soundness:** 3
**Presentation:** 2
**Contribution:** 2
**Rating:** 4
**Confidence:** 3

**Summary:**

This paper addresses the challenge of applying contrastive self-supervised learning (CSSL) to imbalanced datasets, a significant and practical problem given that real-world unlabeled data is often long-tailed.1 The authors propose a novel framework, "More Than Two Views" (MTTV), which departs from the standard paradigm of generating two augmented views from a single image. Instead, MTTV processes two distinct images simultaneously, creating views through a unique combination of an invertible transformation (normalization, denoted $x^n$) and a non-invertible one (standard data augmentation, denoted $x^a$).3 The core idea is to generate a richer set of inter- and intra-image feature comparisons, which is particularly beneficial for learning from the sparse data in tail classes.

**Strengths:**

- Originality of the framework: The core strength of this paper is the conceptual novelty of the MTTV framework. By processing multiple images at once and, most importantly, by combining normalized (invertible) and augmented (non-invertible) views, the authors propose a new and effective way to generate pairs for contrastive learning. This design is well-suited for the low-data regime characteristic of tail classes, as it generates a richer set of training signals from each sample.

**Weaknesses:**

- Overstated Novelty: The paper frames "more than two views" as a key novelty. However, the use of multiple views from a single image is well-established, most notably through the multi-crop augmentation strategy in SwAV (Caron et al., 2020). More recent work, such as "Multiple Positive Views in Self-Supervised Learning", has also explicitly analyzed this setting. The authors fail to differentiate their multi-image approach from these existing single-image, multi-view methods.
- Insufficient Context: Works of SSL for long-tailed recognition are missed. In the specific research direction this paper focuses on—modifying the contrastive learning objective—several highly relevant papers were published in 2023-2025. For instance, "Decoupled Contrastive Learning" (AAAI 2024), BaCon (OpenReview 2024), "Unsupervised Learning of Visual Features by Contrasting Cluster Assignments" et al.
- Experimental Problems:
    - Baseline discrepancy: Under-reporting of the SDCLR baseline on CIFAR10-LT (80.49% in Table 1 vs. 82.00% in the original paper). This calls into question the true magnitude of the improvement and should be corrected.
    - Selective comparison on ImageNet-LT: In Table 3, the comparison on ImageNet-LT is limited to a supervised baseline and MoCo v2. This is an incomplete comparison. For instance, the BCL paper (Zhu et al., CVPR 2022) reports a Top-1 accuracy of 56.7% on ImageNet-LT with a ResNet-50 backbone, which is substantially higher than the 52.90% reported here. This suggests the proposed method is not state-of-the-art on this benchmark, contrary to the paper's claims.
    - Comparing baselines are out of date: The latest baseline is FASSL(2023). The rest highly related works we listed above are not even discussed.
    - Missing Ablation Studies: The paper fails to disentangle the contributions of its three main components: (1) the multi-image setup, (2) the normalized+augmented view strategy, and (3) the thresholded loss. Without ablation studies, it is impossible to determine which of these components is responsible for the performance gains. A rigorous evaluation would require experiments that isolate each component—for example, running MTTV with only augmented views, or running the full framework without the loss thresholds. The absence of these experiments is a major flaw.

**Questions:**

As discussed in the "Weaknesses" section, could you please explain more about the novelty, related work discussion, and experimental problems?

---

> ### Author Response · Authors · 2025-11-20
>
> ### Overstated Novelty
>
> **Reply:**  The novelty comes from the fact that our framework is built from a multi-image, multi-view perspective, whereas SwAV as well as Zhehao Liang et al.al, uses multiple (greater than 2) views from a single image. Moreover, we make use of both invertible (normalized) and non-invertible functions (augmented) for transforming views, whereas any traditional self-supervised framework is designed to make use of only non-invertible functions in the form of augmentations
>
> ### Insufficient Context
>
> **Reply:** "Decoupled Contrastive Learning for Long-tailed recognition" as suggested by the reviewer deals with imbalanced data but in a supervised manner. Similarly, BaCon also deals with a supervised setting for imbalanced data. Though these two works offer insights into dealing with imbalanced data our focus is on self-supervised models for imbalanced data, which we believe is heavily unexplored. We have updated the related work section in which we analyze the differences between the proposed model and the suggested works.Though SWAV uses a self supervised learning approach, our experimental results show the superiority of the proposed method over SWAV.
>
> ### Experimental Problems
>
> **Reply:** We agree to the reviewer comment regarding the  82.0\% accuracy with respect to CIFAR-10-LT as mentioned in the SDCLR (ICML-2021) paper. However we would like to bring it to the notice of the reviewer that the SDCLR paper is talking about linear separability performance rather than linear evaluation accuracy. As far as liner evaluation accuracy is concerned, our model has the upper hand.
>
> **Reply:** Our intention is not to compare the results of supervised learning for imbalanced data versus self supervised approach for imbalanced data. The BCL paper as pointed out by the reviewer falls under the category of supervised approach for imbalanced data but our focus is only on self supervised frameworks. We have removed the supervised baseline from Table.3 to avoid that confusion. Imagenet-LT results are given below.
>
> Model  |          Imagenet (1K) Top-1 Accuracy
> ---------           ----------------------------------------
> Simclr          |              46.81%
> SDCLR        |              46.62%
> MocoV2       |            49.50%
> VicReg         |             38.24%
> Byol             |             43.16%
>
> **Reply:** It should be pointed out that, as far as Self Supervised Learning is concerned, there is very limited work on Imbalanced data. Some exceptions are SDCLR (ICML-2021), BCL-Zhao (ICML-2022), and FASSL(ICCV-workshop- 2023). The comparative results of these methods on CIFAR-10 and CIFAR-100 are described in our paper (Table-1 and Table-2). For ImageNet (Table-3), we have updated the results.
>
> **Reply:** We would like to draw the attention of the reviewer to Section-1. In Figure-2, first we highlight Simclr with two augmented views which is how the  standard Simclr works (denoted as Simclr in the figure). Thereafter results related to Simclr + Normalized + Augmented views (referred to as Simclr + NA in the figure). After that MIMV(earlier MTTV) with both augmented views is shown(MIMV + AA). In the end, we demonstrate MIMV with normalized/augmented views (referred to as MIMV + NA). These results include NT-Xcent loss without threshold.
>
> Model | Linear Evaluation Top-1 Accuracy
> Simclr | 44.49
>
> Simclr + NA | 47.31
>
> MIMV + AA | 48.01
>
> MIMV + NA | 52.84
>
> MIMV + NA + Threshold Loss | 60.18
>
> ### Question- Explanation about Novelty
>
> **Reply:** As far as novelty is concerned, we make the following contributions to justify the multi image multi view perspective.
> 1) We develop an SSL framework wherein multiple images and multiple views (MIMV) are considered as input, which is different from the traditional multiview setup (Figure-3). A more compact representation of different images is achieved through a fusion (sum/concat) operation which helps to better represent features when compared to SIMCLR (Figure-2). 2) We adopt a novel augmentation strategy that includes both normalized (invertible) and augmented (non-invertible) views. In the conventional setup,  augmentation is seen as non-invertible in nature and therefore we tend to lose some information. In our proposed method, the objective is to preserve complete information of one image (invertible) in the form of normalized view as this gives freedom to choose hard augmentation for the other view of the same image. 3) An information theoretic justification based on the information bottleneck (IB) principle is outlined for more optimal representation 4) We introduce a loss function that helps to learn better representations by filtering out extreme features 5) The robustness of the proposed framework is established by applying it to the longtail problem wherein we achieve a new state-of-the-art accuracy (2\% improvement in Cifar10-LT using Resnet-18, 5\% improvement in Cifar100-LT using Resnet-18 and 3\% improvement in Imagenet-LT (1k) using Resnet-50).

---

### Meta-Review · Area_Chair_pTsz · 2026-01-12

**Summary:**

This paper proposes a new method for self-supervised learning; however, after a thorough evaluation of the manuscript and the reviewer comments, I regret that I am unable to provide a recommendation of acceptance for this paper, primarily due to persistent concerns regarding its presentation quality. There is a notable consensus among the review committee regarding this issue, with four reviewers—Reviewer P4oE, Reviewer yEsw, Reviewer FVwq, and Reviewer VRHb—sharing the strong sentiment that the paper is poorly written. The severity of this issue was highlighted by one reviewer who remarked that they considered such a "carelessly prepared submission disrespectful to reviewers," which indicates a significant barrier to understanding the work.

Although I acknowledge and appreciate that the authors have undertaken a major revision in an attempt to improve the writing, I ultimately agree with Reviewer VRHb that the paper, even in its current version, continues to suffer from issues in writing clarity and structure. Furthermore, the experimental section remains somewhat verbose, which hinders the readability of the manuscript. The reviewers have provided a substantial amount of detailed advice for improving this paper, and I believe the authors should express their sincere thanks for this constructive feedback, as it provides a roadmap for necessary improvements.

I am pleased to see that the authors have presented a comprehensive set of responses to the concerns raised by the reviewers. While I recognize that some of these rebuttals have been successful in clarifying certain points, many of the core issues are unfortunately still far from being resolved to a standard that warrants acceptance.

Regarding the technical contributions, I agree with the assessment of Reviewer 8gZQ that the novelty of the work is somewhat overstated. While the authors assert that the novelty stems from the fact that their framework is built from a multi-image, multi-view perspective, I consider this to be a very natural design choice rather than a breakthrough. In fact, almost all contemporary self-supervised learning methods implicitly possess this implementation. Moreover, the authors' specific claim that they make use of both invertible (normalized) and non-invertible (augmented) functions for transforming views appears to be a rather weak novelty that does not sufficiently distinguish this work from the existing state of the art.

**Reviewer Concerns:**

I am pleased to see that the authors have presented a comprehensive set of responses to the concerns raised by the reviewers. While I recognize that some of these rebuttals have been successful in clarifying certain points, many of the core issues are unfortunately still far from being resolved to a standard that warrants acceptance.

**Reviewer Scores:**

I believe that most reviewers are not expected to significantly alter their ratings (e.g., from reject to accept), assuming the evaluation process proceeds fairly and without external influences.

---

### Decision · Program_Chairs · 2026-01-26

Reject